# Longitudinal evidence of technology-enhanced, individualized neuromotor rehabilitation on autonomy, cognition, quality of life, and psychological well-being: Pilot multi-sample study

**Francesco Zanatta**[1], **Patrizia Steca**[1], **Cira Fundarò**[2]*, **Anna Giardini**[3], **Chiara Ferretti**[4], **Giovanni Arbasi**[4], **Roberta Adorni**[1], **Marco D'Addario**[1], **Antonia Pierobon**[5]

**1** Department of Psychology, University of Milano-Bicocca, Milan, Italy, **2** Neurophysiopathology Unit, Istituti Clinici Scientifici Maugeri IRCCS, Montescano Institute, Italy, **3** Information Technology, Istituti Clinici Scientifici Maugeri IRCCS, Pavia, Italy, **4** Neuromotor Rehabilitation Unit, Istituti Clinici Scientifici Maugeri IRCCS, Montescano Institute, Italy, **5** Psychology Unit, Istituti Clinici Scientifici Maugeri IRCCS, Montescano Institute, Italy

* cira.fundaro@icsmaugeri.it

## Abstract

### Background

Over the last decades, neuromotor rehabilitation programs have integrated multidisciplinary approaches with the implementation of emerging technology (e.g., robotics and virtual reality – VR), to effectively target recovery complexity. While this strategy supported patient's physical improvement, little evidence has been reported regarding the widespread effects on non-motor rehabilitation outcomes.

### Methods

A prospective, two-arm, non-randomized study design was adopted to provide pilot feasibility evidence on the multi-domain impact of personalized technology-enhanced neuromotor rehabilitation from convenience sub-samples of patients with stroke, Parkinson's Disease (PD), and osteoarthritis (OA). Technological intervention consisted of the integrated use of robot-assisted and/or VR-based exercises, individualized based on patient's diagnosis and rehabilitation goals. Study outcomes included patient's functional status (autonomy in ADLs, risk of falls), cognition (attention and executive functions, memory, verbal fluency), physical and mental health-related quality of life (HRQoL), and psychological status (anxiety and depression symptoms, and well-being) and were compared to patients participating in standard training only. Rehabilitation experience and technology psychosocial impact were also evaluated. Intra- and intergroup comparisons along with general linear models were statistically

**Data availability statement:** After acceptance, the trial database files will be anonymized and will be made available on ZENODO, a general-purpose open repository developed under the European OpenAIRE program (https://zenodo.org/).

**Funding:** This work was partially supported by the "Ricerca Corrente" funding scheme of the Ministry of Health (Italy). The funder had no role in the study design, data collection and analysis, decision to publish, or preparation of the manuscript.

**Competing interests:** The authors have declared that no competing interests exist.

tested within each sub-sample considered independently over three timepoints (baseline, post-intervention, 6-month follow-up).

## Results

At post-intervention, significant multi-domain intra-group improvements were observed within each sub-sample. Between-group differences were found on ADLs autonomy (stroke and PD; $p < .05$), executive functions (stroke; $p < .01$), anxiety and depression (OA and PD, respectively; $p < .05$), and well-being (stroke and OA; $p < .05$). Interaction effects (time x group) were significant only on well-being variables in stroke ($p = .01$) and OA ($p = .02$), evidencing wider short-term effects of technology-enhanced programs compared to standard training. At 6-month, significant time effects indicating sustained improvements over the three timepoints were estimated on HRQoL within each sub-sample ($p < .05$) and, additionally, on anxiety and depression in stroke ($p = .02$) and OA ($p < .001$). Interaction effects emerged only on physical HRQoL in OA ($p = .02$), along with significant between-group differences on HRQoL and anxiety and depression in OA ($p < .05$) and PD ($p = .01$), respectively.

## Conclusion

Further full-scale trials are warranted to confirm the longitudinal trends observed in this pilot study and to further investigate the potential multi-domain benefits of multi-disciplinary and technology-integrated recovery approaches across different clinical populations.

## Trial registration

ClinicalTrials.gov ID: NCT05399043.

---

## 1. Background

With the progressive shift of paradigm in the definition of disability, the role of multiple and mutual-related biopsychosocial factors in addressing individuals' health and functioning has become increasingly central [1,2]. Contextually, neuromotor rehabilitation programs have shown a growing interest in the adoption of multidisciplinary approaches along with the integration of technology-enabled strategies to effectively target recovery complexity [3,4]. Over the last decades, in particular, emerging technologies like robotics and virtual reality (VR) have been implemented. Thanks to their multipurpose and multimodal application, these technologies have proposed crucial solutions not only to scale up standard interventions but also to potentially respond to a wider spectrum of impairments that result from various acute and chronic conditions, including stroke [5,6], Parkinson's Disease (PD) [7,8], and osteoarthritis (OA) [9,10].

Stroke is one of the leading causes of disability worldwide, with the highest incidences among the neurological diseases and widespread symptoms that often

persist beyond the post-acute phase and require comprehensive and long-term rehabilitation [11]. Post-stroke physical impairments include motor deficits like paralysis, muscles weakness, and impaired coordination that can profoundly limit patients' mobility (e.g., walking, balance) and autonomy in various activities of daily living (ADLs) [12]. Cognitive dysfunctions are also common consequences predicting poor clinical outcomes [13]. Specifically, stroke can severely affect memory, attention, language, and executive functions, often leading to broader detrimental effects on self-care and disease management. Moreover, it was evidenced that stroke can also exacerbate depression and anxiety symptoms, ultimately affecting psychological well-being and increasing patients' limitations to various activities and restrictions to community participation [14].

PD represents another common and complex neurological condition. Physical symptoms typically include movement disorders like tremor at rest, bradykinesia, rigidity, postural instability, and gait dysfunctions [15]. Besides, non-motor impairments can also occur and mainly regard cognition and mental health domains. Cognitive disorders often relate to attentional and executive functions, speech, visuospatial abilities, and memory difficulties, which, in many cases, can significantly predict dementia over time [16]. As for mental health, signs and symptoms often fall into the broad categories of affect (i.e., depression and anxiety), motivation (i.e., impulse control disorders and apathy), and perception and thinking (i.e., psychosis) [17]. Taken together, both motor and non-motor symptoms can cause severe disability, affecting patients' daily life especially as the disease progresses [18].

In conclusion, OA is one of the most frequent musculoskeletal degenerative disorders that, similarly to stroke and PD, induces progressive disability [19]. In particular, it may lead to acute or chronic pain, loss of joint functions, and inflammations, which can make conservative treatments ineffective and cause severe limitations to either basic or complex daily activities. Such condition can exert a broader impact beyond musculoskeletal manifestations, too. As suggested by prior works, persistent pain and physical impairments, for example, can lead to considerable emotional distress, which can translate into increased depression and anxiety symptoms and compound cognitive difficulties [20,21]. Total knee arthroplasty (TKA) and total hip arthroplasty (THA) are considered common surgical procedures performed to alleviate physical symptoms and restore joints functionality. However, despite these procedures have proven to be effective, post-operative rehabilitation process often plays a critical role in ensuring optimal functional recovery [22,23].

Overall, when targeting rehabilitation outcomes in these three pathologies, the adoption of multidimensional and innovative approaches is therefore essential. Recently, advances in motion analysis and wearable sensor-based technologies have further supported technological rehabilitation by enabling objective movement assessment and personalized training planning and implementation [24]. In parallel, emerging biomechanics-informed rehabilitation models have increasingly contributed to optimizing individualized recovery pathways by supporting risk monitoring and functional outcome optimization. However, although the efficacy of technology-enhanced interventions to improve motor impairments has been supported widely [5–10], understanding the short- and long-term broader impact on patients' non-motor characteristics still represents an open challenge [25]. Beyond motor recovery, emerging evidence suggests that technological devices may support multi-domain rehabilitation through several mechanisms, including the delivery of high-intensity and task-specific training, the provision of augmented multisensory feedback, and the facilitation of motor-cognitive integration. In post-stroke rehabilitation, robotic and VR-based interventions have been associated with improved motor learning and engagement, as well as potential secondary benefits on attention, executive functions, and mood [5–6]. In PD, technology-enhanced training has been shown to promote gait automatization, balance control, and dual-task performance, while also positively influencing motivation and perceived well-being [7–8]. Similarly, OA and post-arthroplasty rehabilitation, VR-based interventions may enhance proprioception, balance, functional mobility, while reducing risk and fear of falls and psychological distress [9–10]. Collectively, these advantages support the potential of rehabilitation technologies to restore functional limitations, mitigate secondary complications, and promote recovery across physical, cognitive, and psychological domains. Clearer evidence in this direction not only would extend current knowledge on technology effectiveness to patient's global functioning, but it would also support the added value of providing comprehensive

and technology-enhanced recovery programs. To this end, the integration of personalized approaches represents another essential strategy to optimally target patient's profile. This includes the necessity of concurrently considering patient's medical diagnosis, disability severity, and related rehabilitation objectives, with the final aim of developing and implementing individualized recovery pathways.

Following this line, the present study aimed to provide pilot multi-domain feasibility evidence on the impact of multidisciplinary, individualized, technology-enhanced neuromotor rehabilitation programs. Feasibility was intended as the preliminary assessment of the applicability and potential added value of technology-integrated rehabilitation within real-world clinical settings, as well as its sensitivity in capturing changes across multiple outcome domains. For this purpose, a preliminary longitudinal investigation was conducted in convenience samples of patients drawn from three distinct and representative clinical populations typically undergoing neuromotor rehabilitation, namely stroke (neurological sub-acute), PD (chronic neurodegenerative), and OA (musculoskeletal). The latter was included as a representative musculoskeletal condition commonly undergoing structured neuromotor rehabilitation, particularly in the post-surgical phase. In detail, intervention impact was estimated on patient's functional status, cognitive functioning, psychological profile, and broader health-related quality of life (HRQoL). Perceived rehabilitation experience and technology psychosocial impact were also evaluated at post-intervention phase. Further, longitudinal trajectories (6 months) on the psychological status and HRQoL were finally explored. In line with the pilot and exploratory nature of the study, it was hypothesized that, compared to patients undergoing standard multidisciplinary rehabilitation alone, those participating also in technology-integrated programs would show better short-term multi-domain trends and a more positive rehabilitation experience. It was further exploratorily expected that improvement in psychological well-being and HRQoL would be at least partially maintained at follow-up.

## 2. Materials and methods

The present study is part of a broader project called PHTinRehab Study (*Perception of High Technology in Rehabilitation: a real-life Study on usability, effectiveness, and health-related quality of life*) approved by the Ethics Committee of the Clinical Scientific Institutes Maugeri IRCCS (February 2021, protocol n. 2517CE). The full description of the study protocol was registered in a public clinical trial registry (ClinicalTrials.gov ID: NCT05399043) and published elsewhere [26].

### 2.1. Design, participants, procedures

A pilot prospective, two-arm, open-label, non-randomized study design was adopted.

In a real-world rehabilitation setting, consecutive enrolments were conducted (from September 1st 2021 to July 31th 2022) with patients meeting the following eligibility criteria: 18 years of age or older; diagnosis of stroke, PD or TKA/THA due to OA, requiring rehabilitation intervention; no severe clinical condition (i.e., chronic heart failure at class IV according to the New York Heart Association classification – NYHA-IV, ischemic heart disease at class IV according to Canadian Cardiovascular Society classification – CCS-IV, neoplastic diseases or acute respiratory disease); no cognitive impairment (screened with the Montreal Cognitive Assessment – MoCA>15.5 [27]); no language disorders (e.g., aphasia); no severe mental health condition or psychiatric disorders (assessed according to the Diagnostic and Statistical Manual of Mental Disorders – DSM-V-TR) that can potentially compromise participation in the study and reliability of collected data, and no prior exposure structured exposure to the rehabilitation technologies implemented in this study. All the patients involved were given the informed written consent and asked to sign it to join the study. Their participation was voluntary and did not affect the healthcare process. The entire study was conducted in accordance with the Declaration of Helsinki and all relevant guidelines and regulations covering respect for the rights and dignity of participants.

At hospital admission (or after clinical stabilization), patients were screened for eligibility and asked to participate. Enrolment was carried out by a multidisciplinary panel consisting of physical and rehabilitation medicine specialists, neurologists, psychologists, and researchers. Included patients underwent a comprehensive baseline evaluation of the study

outcomes, which was replicated at post-intervention phase (after 4 weeks). Here, perceived rehabilitation experience and technology psychosocial impact were further evaluated. Finally, follow-up telephone evaluations at 6 months after the intervention were conducted, specifically on patient-reported autonomy in ADLs, psychological status, and HRQoL.

## 2.2. Intervention

Patients were assigned to the two study arms, depending on their rehabilitation program and objectives (Fig 1). These were defined following the routine multidisciplinary clinical practice of the Institution where the study was carried out [28]. In particular, each patient's recovery pathway was designed individually based on the integration between the International Classification of Diseases (ICD) and the International Classification of Functioning, Disability, and Health (ICF)

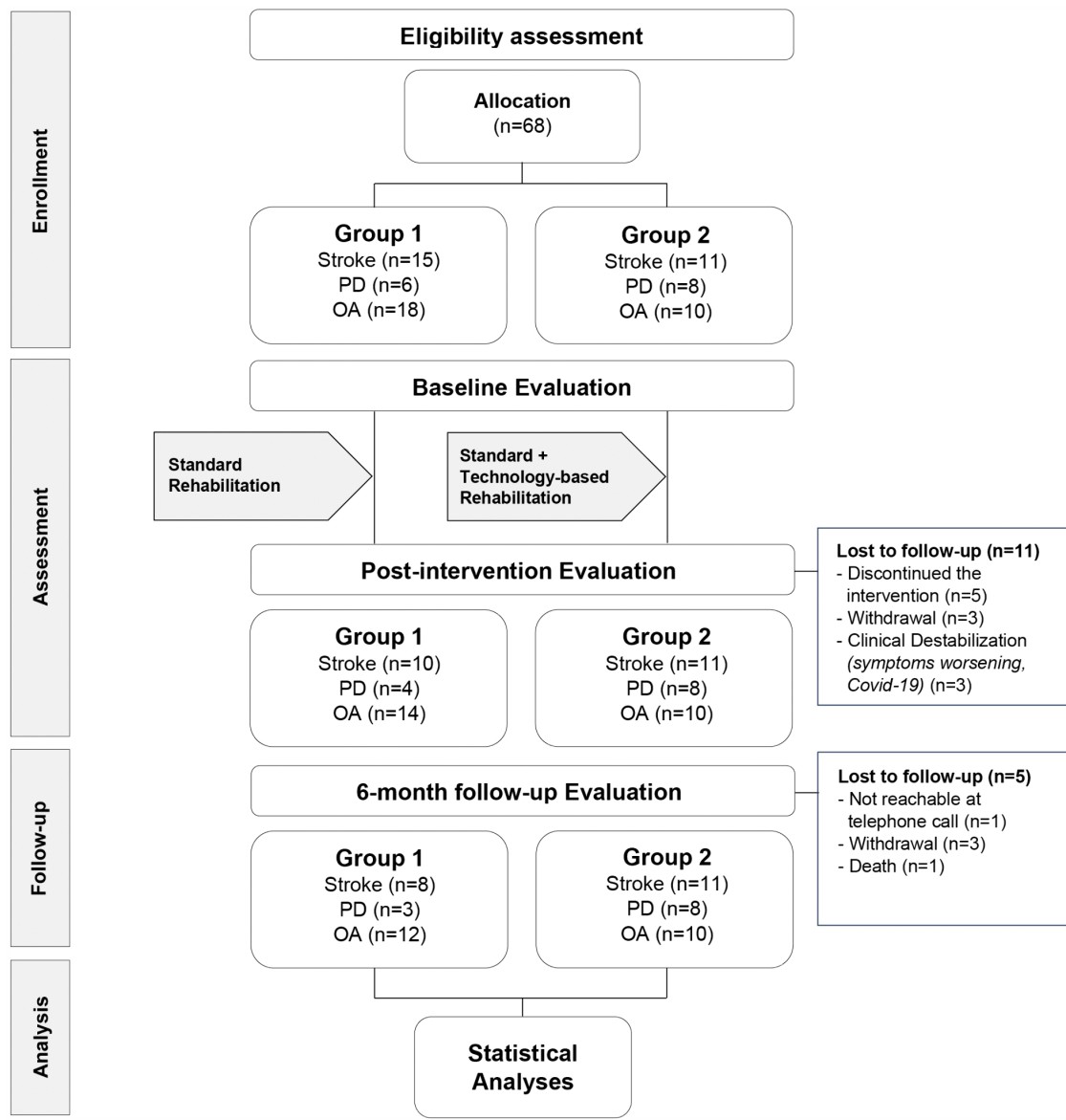

**Fig 1. Flow diagram of the study.**

models [29]. Therefore, based on their rehabilitation needs, patients could have participated in further activities besides physical therapy, namely occupational therapy, speech therapy, cognitive stimulation, and/or psychological support. Accordingly, rehabilitation procedures followed a biopsychosocial approach and were adapted according to one's specific diagnosis, disability severity, and recovery objectives, ultimately delivering personalized rehabilitation projects and programs. Although participants did not undergo randomization, the approximate balance in sample sizes across study arms and study groups resulted from Institution's routine effort to optimize resource allocation and ensure equitable access to care, reflecting real-world clinical practice.

All participants underwent standard treatment which consisted of two daily 1-hour sessions of physical therapy over 4 weeks. These included rehabilitation activities like range of motion exercises (passive, active assisted, and active), balance, strength and proprioceptive training, progressive resistive exercises, and aerobic conditioning. Only the patients assigned to the second study group arm underwent technology-based training. For these, part of the duration (which depended on rehabilitation objectives) of the whole intervention consisted of the use of exoskeletal (Lokomat®, Armeo-Spring® – Hocoma AG, Switzerland) and/or VR technology (ProKin, D-Wall, Walker View – TecnoBody SRL, Italy), resulting in the same amount of therapy for all participants. Also, intervention intensity and workload were routinely monitored through functional performance observations, device-generated training parameters (e.g., assistance levels, task difficulty, and progression criteria) to ensure comparable therapeutic exposure across groups. Technology-enhanced rehabilitation procedures, including eligibility criteria for assignment and implementation, followed appropriate rehabilitation technology-specific guidelines and recommendations [30–34].

## 2.3. Data collection

Following preliminary collection of socio-demographic and clinical data (i.e., age, gender, marital status, living condition, education, Body Mass Index – BMI, comorbidities, and risk factors), all participants were evaluated through a multidimensional battery of standardized scales and questionnaires, whose full description, including their psychometric properties, is provided in the original study protocol published elsewhere [26].

Functional status evaluation included an assessment of the autonomy in the ADLs through the Modified Barthel Index (MBI) [35] and the Functional Independence Measure (FIM) [36]. Risk of falls was also evaluated through the Morse Fall Scale [37]. At 6-month follow-up, the Basic Activities of Daily Living (BADL) [38] and the Instrumental Activities of Daily Living (IADL) [39] were also administered.

Cognitive functioning was assessed through a comprehensive battery of neuropsychological tests. These included the Montreal Cognitive Assessment (MoCA) [40], which assessed global cognition (permission to use for research purposes was requested and training certification to administer the test was obtained). Attention, executive functions, and linguistic skills were further evaluated through the oral version of the Symbol Digit Modalities Test (SDMT) [41], the Trail Making Test (TMT-A and TMT-B) [42], the shortened version of the Stroop Colour Word test [43], the Frontal Assessment Battery [44], and the phonemic verbal fluency test [45]. All tests were adjusted for participants' age and education to generate standardized scores comparable to normative data available for each test.

HRQoL evaluation was carried out through the Short-Form Health Survey-12 (SF-12) [46], which allowed to obtain composite scores for physical (PCS) and mental (MCS) health status, and the visual analogue scale of the Euro-QoL (EQ-VAS) [47] for overall health (permission to use the scale was requested and obtained). Anxiety and depression symptoms were also evaluated with the General Anxiety Disorder-7 (GAD-7) [48] and the Patient Health Questionnaire-9 (PHQ-9) [49], respectively. Moreover, the Satisfaction-Profile (SAT-P) [50] was administered as a measure of well-being to specifically evaluate patient's perceived satisfaction in the last month with mood, resistance to physical fatigue, and mental efficiency. At post-intervention, rehabilitation experience was evaluated with the Client-Centred Rehabilitation Questionnaire (CCRQ) [51], while the psychosocial impact of the technology implemented was assessed with the Psychosocial Impact of Assistive Device Scale (PIADS) [52], which evaluates the perceived benefits of assistive technologies

on functional ability and independence (i.e., ability subscale), motivation and ability to adapt to changes (i.e., adaptability subscale), and emotional well-being and self-confidence (i.e., self-esteem subscale). At 6-month follow-up, the SF-12 and the EQ-VAS were administered again, along with an assessment of anxiety and depression symptoms through a short-ened version of the Patient Health Questionnaire (PHQ-4) [53].

## 2.4. Data analysis

Descriptive statistics on all the study variables were calculated. Mean and standard deviations for continuous variables and percentages for categorical variables were reported. For each sub-sample, between-group statistical differences were then calculated to check if the two study arms were similar at the baseline. Intervention characteristics and rehabilitation experience were also analysed from all participants. Of those also undergoing technology-based training, the number of sessions and total minutes of devices use were extracted and described along with the perceived technology psychosocial impact. Within- and between-groups pre-post intervention differences were then calculated on the study outcomes (i.e., functional status, cognition, HRQoL, and psychological status). Of these, delta values were also generated, by calculating the difference between post-intervention and baseline mean scores, and then compared between the two study arms. In addition, repeated-measures ANOVA models were tested to estimate the main (time) and interaction (time x group) effects emerged from the intervention on all the study outcomes. Further models were finally tested including also the outcomes evaluated at 6-month follow-up (i.e., HRQoL, anxiety and depression symptoms). For each of them, assumption of sphericity was assessed through Mauchly's test. In cases of assumption violation, appropriate adjustments were applied. For all pairwise comparisons, Bonferroni correction was used (see supplementary material for further details). Partial eta squared ($\eta2p$) coefficients as indicator of effect size were generated along with achieved statistical power. All the statisti-cal analyses were replicated within each study sample involved (i.e., stroke, PD, and OA) and described independently. The Statistical Package for the Social Sciences (SPSS, v.29.0) was used and a p-value ≤ .05 was considered statistically significant.

## 3. Results

### 3.1. Sub-samples characteristics and intervention

Baseline socio-demographic and clinical characteristics of the study samples are presented in Table 1. For each of them, no significant between-group differences emerged, meaning that study participants were similar before the intervention.

Table 2 shows the intervention characteristics, including the results on the perceived rehabilitation overall experience and technology psychosocial impact. Besides standard physical therapy, most of the patients participated in parallel reha-bilitation activities, except for those following TKA/THA. As for the use of technology, patients with stroke or PD used VR or exoskeletal devices, while those following TKA/THA underwent VR-based training only. As already mentioned, tech-nology assignment and implementation strictly depended on patient's individualized rehabilitation objectives and project. In general, however, robotic devices were considered more suitable for patients with stroke or PD, while the use of VR devices, specifically the ProKin platform was preferred for patients following TKA or THA given its potential to provide enhanced proprioceptive, balance, and sensory-motor training, which is essential for joints post-surgery recovery.

All patients reported satisfactory levels of perceived rehabilitation experience. Specifically, significant between-group differences were found among patients with PD in emotional support ($p = .01$) and physical comfort ($p = .04$) subscales, and among patients following TKA/THA in outcome evaluation subscale ($p = .03$), where who used technology reported higher mean scores. No further significant differences were estimated across the three study samples, although in general higher scores were observed in patients undergoing technology-based rehabilitation in all CCRQ subscales.

In conclusion, from the evaluation of the psychosocial impact of technology, positive mean scores were observed for each subscale of the PIADS. In detail, participants reported that the use of technological devices positively affected

**Table 1. Baseline socio-demographic and clinical characteristics of the study samples.**

| Variables | Stroke | | | PD | | | OA | | |
|---|---|---|---|---|---|---|---|---|---|
| | Group 1 (n=10) | Group 2 (n=11) | p | Group 1 (n=4) | Group 2 (n=8) | p | Group 1 (n=14) | Group 2 (n=10) | p |
| Age, Mean±SD | 68.5±9.6 | 66.0±11.1 | .705 | 75.8±10.9 | 73.9±9.9 | .683 | 70.3±8.9 | 69.9±8.7 | .752 |
| Gender, n(%) | | | .835 | | | .221 | | | .124 |
| *Male* | 5(50.0) | 5(45.5) | | 1(25.0) | 5(62.5) | | 4(28.6) | 6(60.0) | |
| *Female* | 5(50.0) | 6(54.5) | | 3(75.0) | 3(37.5) | | 10(71.4) | 4(40.0) | |
| Marital Status, n(%) | | | .256 | | | .679 | | | .327 |
| *Single/Separated/widowed* | 3(30.0) | 6(54.5) | | 2(50.0) | 5(62.5) | | 7(50.0) | 3(30.0) | |
| *Married* | 7(70.0) | 5(45.5) | | 2(50.0) | 3(37.5) | | 7(50.0) | 7(70.0) | |
| Living condition, n(%) | | | .407 | | | .211 | | | .134 |
| *Alone* | 2(20.0) | 4(36.4) | | 1(25.0) | 5(62.5) | | 7(50.0) | 2(20.0) | |
| *With others* | 8(80.0) | 7(63.3) | | 3(75.0) | 3(37.5) | | 7(50.0) | 8(80.0) | |
| Education, n(%) | | | .531 | | | .638 | | | .605 |
| *None or primary* | 3(30.0) | 4(36.4) | | 2(50.0) | 2(25.0) | | 6(42.9) | 3(30.0) | |
| *Middle school* | 4(40.0) | 2(18.2) | | 1(25.0) | 4(50.0) | | 5(35.7) | 3(30.0) | |
| *High school or higher* | 3(30.0) | 5(45.5) | | 1(25.0) | 2(25.0) | | 3(21.4) | 4(40.0) | |
| BMI, Mean±SD | 26.2±4.4 | 24.4±3.2 | .654 | 28.2±7.4 | 24.3±4.7 | .461 | 28.3±6.9 | 28.2±3.2 | .666 |
| Comorbidity, n(%)* | | | .694 | | | .223 | | | .382 |
| *None* | 3(30.0) | 3(27.3) | | – | 4(50.0) | | 9(64.3) | 7(70.0) | |
| *One* | 2(20.0) | 4(36.4) | | 1(25.0) | 1(12.5) | | 5(35.7) | 2(20.0) | |
| *Two or more* | 5(50.0) | 4(36.4) | | 3(75.0) | 3(37.5) | | – | 1(10.0) | |
| Risk Factors, n(%)° | | | .525 | | | .350 | | | .533 |
| *None* | 1(10.0) | – | | – | 3(37.5) | | 6(42.9) | 5(50.0) | |
| *One* | 5(50.0) | 7(63.6) | | 2(50.0) | 2(25.0) | | 4(28.6) | 4(40.0) | |
| *Two or more* | 4(40.0) | 4(36.4) | | 2(50.0) | 3(37.5) | | 4(28.6) | 1(10.0) | |

Notes. Group 1, Standard training; Group 2, Technology-enhanced training. p-values are from chi-squared test and Mann-Whitney test for categorical and continuous variables, respectively. BMI, Body Mass Index; SD, Standard deviation

*Comorbidities include diabetes, polyneuropathy, osteoporosis, polymyalgia rheumatica, obstructive sleep apnea, chronic obstructive pulmonary disease, atrial fibrillation, and coronary heart disease.

°Risk factors include smoking behaviour, dyslipidaemia, arterial hypertension, hyperuricemia, diseases familiarity, past clinical events.

their capability to perform actions and activities and to face daily tasks (ability subscale), their willingness to cope with new experiences and challenges and to adapt to different settings (adaptability subscale), as well as their mood, self-confidence, and emotions (self-esteem subscale).

### 3.2. Intervention effectiveness

Pre- post-intervention changes of the study participants are showed in Table 3. Multi-domain impacts of the rehabilitation programs were observed on multiple study outcomes, along with significant main and interaction effects. Intervention effects within each study sample over the study phases are presented. Further details on technology-based intervention delivered (including number of training sessions and total minutes of devices use), outcomes mean scores and the analyses conducted, including those at 6-month follow-up, are provided as supplementary material (S1 Appendix).

 **3.2.1. Stroke.** Post-intervention improvements were observed within both study arms in all the outcomes evaluated.
 Significant changes were found in the FIM mean scores within both the Group 1 (FIM Total: $p=.007$; FIM Motor: $p=.007$) and Group 2 (FIM Total: $p=.01$; FIM Motor: $p=.01$), while only within the Group 2 significant improvements were

**Table 2. Intervention characteristics, and rehabilitation experience (CCRQ) and technology psychosocial impact (PIADS) mean scores of the study samples.**

| Variable | Stroke | | | PD | | | OA | | |
|---|---|---|---|---|---|---|---|---|---|
| | Group 1 (n = 10) | Group 2 (n = 11) | p | Group 1 (n = 4) | Group 2 (n = 8) | p | Group 1 (n = 14) | Group 2 (n = 10) | p |
| Parallel Rehabilitation activities, n(%) | 8(80.0) | 8(72.2) | .696 | 2(50.0) | 5(62.5) | .679 | – | – | – |
| CCRQ, Mean±SD * | | | | | | | | | |
| Client Participation (6–30) | 26.7±4.1 | 28.3±2.1 | .739 | 19.5±7.5 | 27.4±1.9 | .073 | 25.1±3.9 | 28.4±1.8 | .084 |
| Client Centred Education (5–25) | 20.9±3.9 | 20.6±1.2 | .222 | 16.0±3.4 | 20.3±3.0 | .073 | 20.2±4.1 | 22.3±3.1 | .284 |
| Outcome Evaluation (4–20) | 17.6±1.3 | 18.3±1.7 | .370 | 14.0±4.6 | 17.0±3.2 | .283 | 16.4±2.9 | 18.8±1.4 | **.031** |
| Family Involvement (5–25) | 21.4±3.6 | 21.4±5.4 | .779 | 14.3±4.0 | 22.6±3.3 | .071 | 21.4±3.6 | 21.5±4.9 | .889 |
| Emotional Support (5–25) | 17.5±4.6 | 19.6±1.2 | .251 | 14.0±6.1 | 19.3±0.9 | **.012** | 17.8±2.7 | 19.2±1.2 | .292 |
| Physical Comfort (4–20) | 18.1±2.6 | 19.1±1.3 | .720 | 14.5±5.1 | 18.6±1.4 | **.048** | 17.5±2.8 | 19.3±1.1 | .131 |
| Continuity/Coordination (5–25) | 18.1±2.7 | 18.6±2.2 | .809 | 13.5±3.5 | 17.3±2.2 | .214 | 18.2±2.5 | 20.1±1.4 | .067 |
| PIADS, Mean±SD (range: −3/+3) | | | | | | | | | |
| Ability | – | 1.7±0.9 | | – | 1.5±0.8 | | – | 1.0±0.7 | |
| Adaptability | – | 1.8±1.0 | | – | 1.7±0.9 | | – | 0.9±0.9 | |
| Self-esteem | – | 1.6±0.8 | | – | 1.4±0.7 | | – | 0.9±0.5 | |

Notes. Group 1, Standard training; Group 2, Technology-enhanced training. p-values are from chi-squared test and Mann-Whitney test for categorical and continuous variables, respectively.

Parallel Rehabilitation activities included occupational therapy, speech therapy, psychological support, and cognitive stimulation.

*Range values are reported for each CCRQ subscale.

CCRQ, Client-Centred Rehabilitation Questionnaire; PIADS, Psychosocial Impact of Assistive Devices Scale.

estimated in the MBI scores ($p = .02$). Significant main effects were estimated in the FIM total scores ($p < .001$, $\eta^2$p = .525) and FIM motor subscale ($p < .001$, $\eta^2$p = .561).

Improvements in cognitive outcomes were also observed. Significant changes were estimated within the Group 1 in the SDMT ($p = .02$) and Stroop error interference ($p = .03$) scores, while the Group 2 significantly improved MoCA ($p = .03$), SDMT ($p = .02$), TMT-B ($p < .05$), and FAB ($p = .02$) scores. A significant main effect was also estimated on MoCA ($p < .01$, $\eta^2$p = .335), SDMT ($p < .001$, $\eta^2$p = .478), TMT-B ($p = .04$, $\eta^2$p=219), and FAB ($p < .01$, $\eta^2$p = .311) scores, but no interaction effects or between-group differences were found.

As for HRQoL and psychological outcomes, significant improvements were found within the Group 1 in EQ-VAS ($p = .04$), SF-12 PCS ($p = .04$), SF-12 MCS ($p = .01$), GAD-7 ($p = .02$), and SAT-P resistance to physical fatigue ($p = .01$) scores. Within the Group 2, significant changes were estimated in EQ-VAS ($p = .02$), SF-12 MCS ($p = .02$), GAD-7 ($p = .02$), PHQ-9 ($p = .02$), SAT-P mood ($p < .01$) and resistance to physical fatigue ($p = .02$) scores. Significant main effects were also estimated on EQ-VAS ($p < .01$, $\eta^2$p = .425), SF-12 PCS ($p = .01$, $\eta^2$p = .301), SF-12 MCS ($p < .001$, $\eta^2$p = .593), GAD-7 ($p < .001$, $\eta^2$p = .454), PHQ-9 ($p < .01$, $\eta^2$p = .357), and SAT-P mood ($p < .001$, $\eta^2$p = .490) and resistance to physical fatigue ($p < .001$, $\eta^2$p = .542) scores. Moreover, a significant interaction effect between time and group was estimated on SAT-P mood ($p = .02$, $\eta^2$p = .275) scores, along with significant between-group differences ($p = .01$) with the Group 2 showing wider improvements.

Fig 2 shows the results from the longitudinal analyses. For all measures, no between-group differences were found at any timepoint and no significant interactions between time and group were observed. However, significant main effects were estimated on the EQ-VAS ($p < .01$, $\eta^2$p = .244), SF-12 PCS ($p = .01$, $\eta^2$p = .228), SF-12 MCS ($p < .001$, $\eta^2$p = .455), and PHQ-4 ($p = .02$, $\eta^2$p = .202) over the three study timepoints.

**Table 3. Intervention effects on functional, cognitive, HRQoL and psychological outcomes for each study sample.**

| | Stroke | | | | | | | PD | | | | | | | OA | | | | | | |
|---|---|---|---|---|---|---|---|---|---|---|---|---|---|---|---|---|---|---|---|---|---|
| | Group 1 Δ | Group 2 Δ | $p_{(\Delta)}$ | ANOVA_Time p | ANOVA_Time $\eta^2_p$ | ANOVA_Interaction p | ANOVA_Interaction $\eta^2_p$ | Group 1 Δ | Group 2 Δ | $p_{(\Delta)}$ | ANOVA_Time p | ANOVA_Time $\eta^2_p$ | ANOVA_Interaction p | ANOVA_Interaction $\eta^2_p$ | Group 1 Δ | Group 2 Δ | $p_{(\Delta)}$ | ANOVA_Time p | ANOVA_Time $\eta^2_p$ | ANOVA_Interaction p | ANOVA_Interaction $\eta^2_p$ |
| **Functional** | | | | | | | | | | | | | | | | | | | | | |
| MBI | 9.8(34.5) | **9.0(11.3)** | .468* | .103 | .134 | .943 | .000 | 8.0(10.2) | 13.4(19.8) | 1.00 | .074 | .285 | .626 | .025 | 16.3(19.7) | 9.0(8.5) | .508 | **.001** | .394 | .287 | .051 |
| MFS | −1.0(7.4) | −5.0(8.8) | .353 | .116 | .131 | .286 | .063 | −2.5(5.0) | 0.6(6.8) | .570 | .637 | .023 | .437 | .062 | −1.8(6.7) | −4.0(9.7) | 1.00 | .096 | .121 | .513 | .020 |
| **FIM** | | | | | | | | | | | | | | | | | | | | | |
| *Motor* | **17.1(17.6)** | **14.1(9.2)** | .897* | **<.001** | .561 | .672 | .011 | 15.8(6.3) | 14.3(1.6) | .648* | **<.001** | .945 | .560 | .039 | 16.3(12.1) | 14.6(5.3) | .794 | **<.001** | .717 | .690 | .008 |
| *Cognitive* | 1.1(3.1) | 0.5(1.3) | .897 | .198 | .101 | .621 | .016 | 0.0(0.0) | 0.6(0.8) | .315 | .190 | .183 | .190 | .183 | 0.0(0.0) | 0.1(0.3) | .695 | .238 | .069 | .238 | .069 |
| *Total* | **18.2(20.5)** | **14.5(8.8)** | .696* | **<.001** | .525 | .641 | .014 | 15.8(6.3) | 14.9(1.6) | .412 | **<.001** | .947 | .720 | .015 | 16.3(12.1) | 14.7(5.2) | .744 | **<.001** | .720 | .708 | .007 |
| **Cognitive**[1] | | | | | | | | | | | | | | | | | | | | | |
| MoCA | 1.8(3.1) | **2.4(2.9)** | .512 | **.006** | .335 | .661 | .010 | 3.4(2.6) | 0.8(2.2) | .073 | **.015** | .461 | .098 | .249 | **1.6(2.5)** | 1.4(3.7) | .877 | **.032** | .202 | .859 | .002 |
| SDMT | **6.0(6.3)** | **4.0(4.6)** | .557 | **<.001** | .478 | .407 | .036 | 3.0(2.6) | 6.1(7.4) | .497 | .073 | .314 | .505 | .051 | 1.5(3.4) | **3.9(4.6)** | .267 | **.005** | .332 | .186 | .086 |
| TMT-A | −5.5(14.8) | −7.3(25.1) | .863 | .177 | .094 | .840 | .002 | 18.1(20.6) | −3.1(16.4) | .133 | .233 | .154 | .105 | .265 | **−9.2(12.9)** | −1.3(7.6) | .212 | **.048** | .182 | .130 | .111 |
| TMT-B | −12.2(42.9) | **−76.4(114.4)** | .211* | **.043** | .219 | .132 | .129 | 12.3(52.7) | −26.6(64.1) | .376 | .740 | .013 | .376 | .088 | 17.2(71.3) | 12.0(43.9) | .815 | .308 | .052 | .854 | .002 |
| *Stroop Errors* | **−1.5(1.7)** | −0.9(3.2) | .282 | .057 | .177 | .601 | .015 | 0.3(0.6) | −1.1(3.7) | .497 | .729 | .014 | .526 | .046 | −2.8(6.7) | −0.9(2.7) | .714 | .146 | .103 | .453 | .028 |
| *Stroop Time* | −4.3(7.9) | −0.4(10.9) | .349 | .277 | .062 | .359 | .044 | −5.1(22.1) | −3.9(11.6) | 1.00 | .358 | .085 | .907 | .002 | 2.5(13.2) | −5.9(8.7) | .127 | .521 | .021 | .125 | .114 |
| FAB | 1.9(3.5) | **1.7(1.9)** | .809 | **.009** | .311 | .849 | .002 | −0.2(1.2) | 0.9(1.4) | .194 | .437 | .069 | .285 | .126 | 0.8(2.2) | 1.8(2.9) | .557 | **.023** | .224 | .379 | .037 |
| Verbal fluency | −0.8(5.2) | 2.4(4.3) | .132 | .461 | .029 | .145 | .109 | −0.7(1.5) | 2.0(9.5) | .376 | .820 | .006 | .650 | .024 | −1.1(6.5) | −1.2(8.2) | .781 | .466 | .026 | .956 | .000 |
| **HRQoL and Psychological** | | | | | | | | | | | | | | | | | | | | | |
| EQ-VAS | **25.5(35.4)** | **22.9(23.1)** | .863 | **.001** | .425 | .843 | .002 | 0.0(21.6) | **18.1(20.2)** | .214* | .181 | .171 | .181 | .171 | 7.5(12.6) | **8.5(12.3)** | .752 | **.005** | .304 | .848 | .002 |
| **SF-12** | | | | | | | | | | | | | | | | | | | | | |
| PCS | **7.7(11.9)** | 4.9(8.3) | .557 | **.010** | .301 | .549 | .019 | 5.3(15.1) | 5.5(9.8) | .368 | .114 | .231 | .870 | .003 | **8.9(5.7)** | **7.2(8.1)** | .172 | **<.001** | .647 | .525 | .019 |
| MCS | **14.2(11.6)** | **11.9(11.1)** | .605 | **<.001** | .593 | .636 | .012 | −2.7(12.4) | 6.4(8.7) | .808 | .679 | .018 | .238 | .136 | −0.7(5.6) | 7.2(6.7) | .709* | .302 | .048 | .133 | .100 |
| GAD-7 | **−4.4(4.8)** | **−4.9(5.8)** | 1.00 | **<.001** | .454 | .830 | .002 | −2.3(1.3) | −4.4(4.3) | .570 | **.037** | .367 | .738 | .012 | −1.3(2.9) | **−4.5(5.5)** | .138* | **.003** | .336 | .077 | .135 |
| PHQ-9 | −4.6(6.8) | **−4.1(5.4)** | 1.00 | **.004** | .357 | .851 | .002 | −2.3(2.5) | **−3.0(4.2)** | .570* | **.018** | .444 | .386 | .076 | −1.8(4.1) | −2.6(4.0) | .709 | **.016** | .237 | .632 | .011 |
| **SAT-P** | | | | | | | | | | | | | | | | | | | | | |
| *Mood* | 7.9(29.3) | **34.6(14.7)** | **.013** | **<.001** | .490 | **.015** | .275 | −7.0(38.9) | **30.9(25.9)** | .109 | .229 | .141 | .070 | .292 | 10.1(23.3) | 12.0(16.0) | .886* | **.017** | .233 | .823 | .002 |
| *RPF* | **37.8(33.0)** | **29.1(31.6)** | .426* | **<.001** | .542 | .544 | .020 | 26.3(55.3) | **27.8(34.0)** | 1.00 | .060 | .310 | .954 | .000 | 13.0(30.4) | **20.1(24.6)** | .546* | **.010** | .268 | .549 | .017 |
| *ME* | 7.4(23.6) | 13.5(28.3) | .349 | .083 | .150 | .597 | .015 | 21.5(20.6) | **15.5(17.1)** | .808 | **.008** | .525 | .602 | .028 | −6.7(18.9) | **10.1(14.9)** | .019* | .642 | .010 | **.029** | .200 |

Notes. Group 1, Standard training; Group 2, Technology-enhanced training. Values in bold are statistically significant (*Wilcoxon signed-rank test*).

1 raw scores are adjusted for age and education.

Δ, difference (delta) between pre- and post-intervention values. Values are means and standard deviations (SD).

$p_{(\Delta)}$, significance from between-group comparisons of delta values (*Mann-Whitney test*).

* Significant between-group differences at post-intervention (Group 2 superiority).

$\eta^2_p$, partial eta squared coefficient.

MBI, Modified Barthel Index; MFS, Morse Fall Scale; FIM, Functional Independence Measure; MoCA, Montreal Cognitive Assessment; SDMT, Symbol Digit Modalities Test; TMT, Trail Making Test; FAB, Frontal Assessment Battery; EQ-VAS, EuroQol Visual-Analogue Scale; GAD-7, General Anxiety Disorder-7; PHQ-9, Patient Health Questionnaire-9; SAT-P, Satisfaction-Profile (RPF, Resistance to Physical Fatigue; ME, Mental Efficiency).

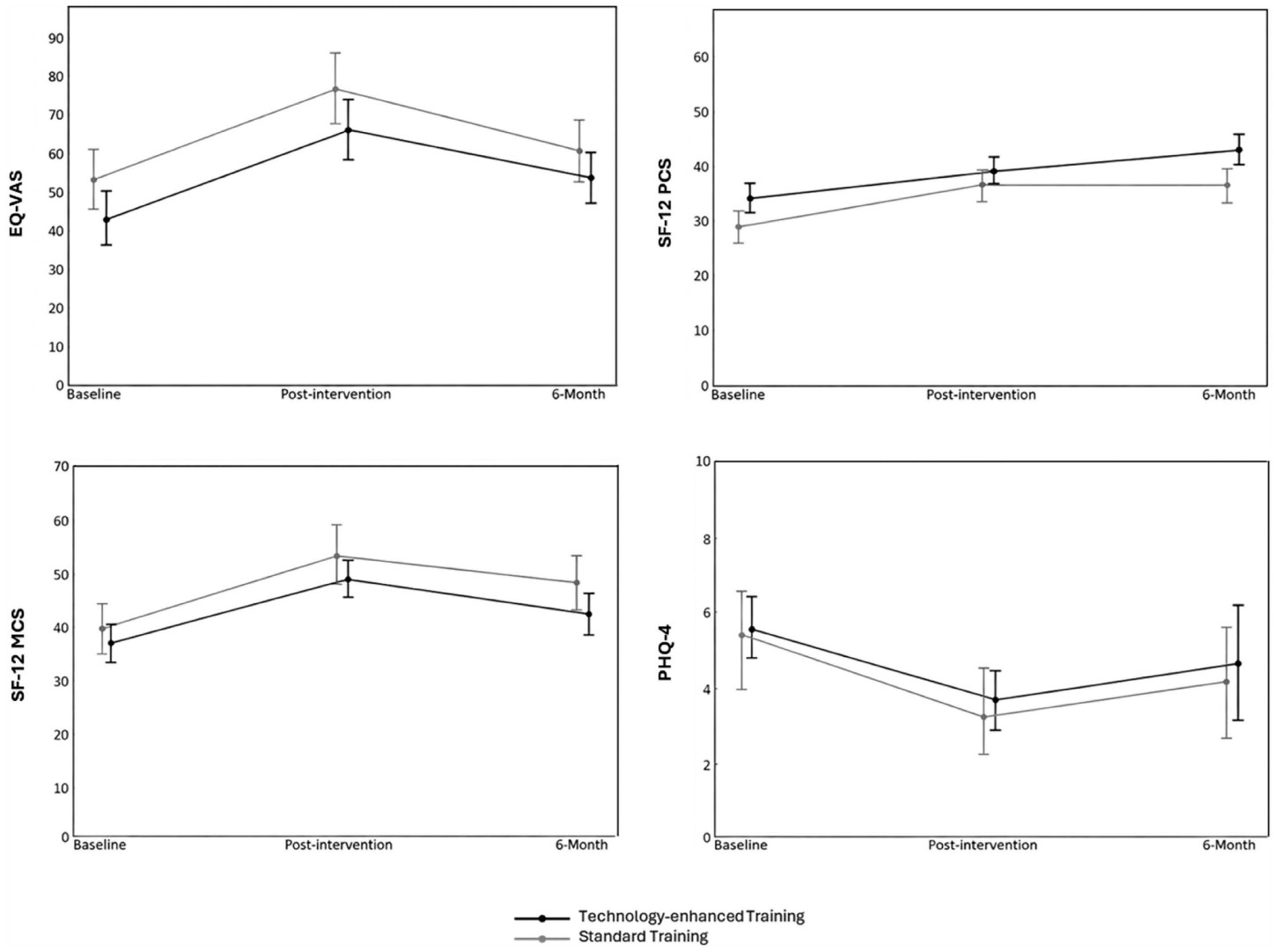

**Fig 2. HRQoL, anxiety and depression mean scores at baseline, post-intervention, and 6-month in patients with stroke.**

### 3.2.2. Parkinson's disease (PD).
Multi-domain improvements were also observed among patients with PD.

Significant changes in the MBI ($p = .03$), FIM motor ($p = .02$) and FIM total ($p = .02$) scores were estimated within the Group 2 only. At post-intervention, a significant between-group difference in FIM motor subscale ($p = .04$) was found, with the Group 2 reporting higher mean scores than the Group 1. Moreover, a significant main effect was observed on FIM motor ($p < .001$, $\eta^2 p = .945$) and FIM total ($p < .001$, $\eta^2 p = .947$) scores.

As for cognition, no statistically significant changes or between-group comparisons were estimated, although clinical improvements can be observed in all the neuropsychological tests performed within both groups. Only a significant main effect of time on MoCA scores ($p = .02$, $\eta^2 p = .461$) emerged.

Regarding HRQoL and psychological outcomes, significant improvements were found within the Group 2 exclusively, specifically in EQ-VAS ($p = .04$), PHQ-9 ($p = .03$), and SAT-P mood ($p = .01$), resistance to physical fatigue ($p = .01$), and mental efficiency ($p = .03$) scores. Significant between-group differences were also observed, at post-intervention, in

EQ-VAS ($p < .05$) and PHQ-9 ($p = .02$) scores, with the Group 2 showing better HRQoL and lower depression symptoms than the Group 1. A significant main effect was then estimated on GAD-7 ($p = .04$, $\eta^2p = .367$), PHQ-9 ($p = .02$, $\eta^2p = .444$), and SAT-P mental efficiency ($p < .01$, $\eta^2p = .525$) scores. For all measures, no significant interaction effects or pre-post intervention between-group differences were observed.

Fig 3 shows the mean scores of HRQoL variables and anxiety and depression symptoms over the three study time-points. A significant main effect was estimated on on SF-12 PCS ($p = .02$, $\eta^2p = .347$) scores, but no interaction effects were found. Particularly for PHQ-4 scores, a significant difference between baseline and 6-month follow-up scores was observed within the Group 2 ($p < .05$), which also reported significantly lower scores than the Group 1 at final follow-up evaluation ($p = .01$).

**3.2.3. Osteoarthritis (OA).** Widespread effects were finally observed among patients following TKA/THA, too.

Significant changes within both study groups were found in MBI (Group 1: $p = .003$; Group 2: $p = .02$), FIM motor (Group 1: $p = .001$; Group 2: $p = .008$), and FIM total (Group 1: $p = .001$; Group 2: $p = .008$) scores, along with significant main

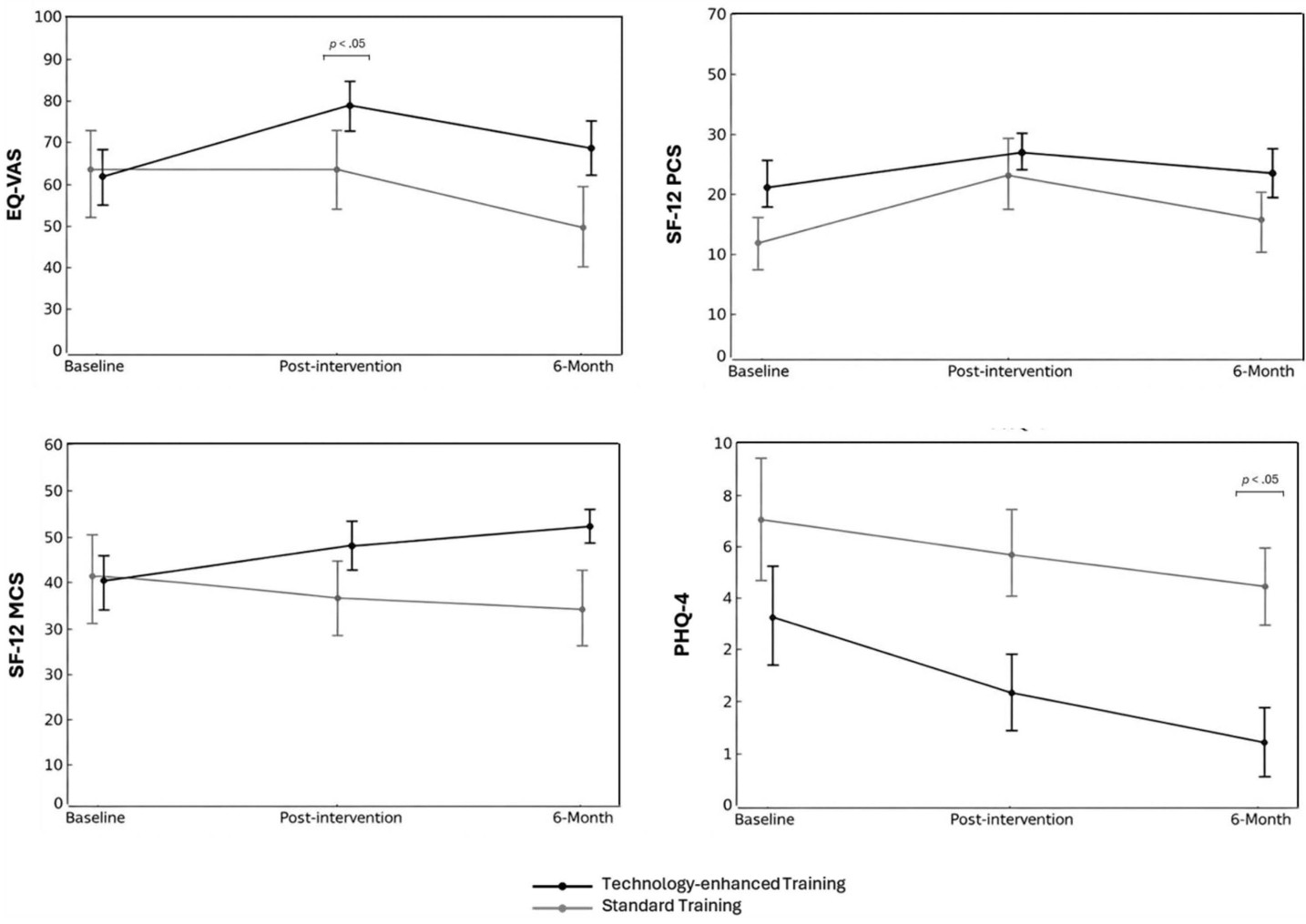

**Fig 3. HRQoL, anxiety and depression mean scores at baseline, post-intervention, and 6-month in patients with PD.**

effects of time (MBI: $p = .001$, $\eta^2 p = .394$; FIM motor: $p < .001$, $\eta^2 p = .717$; FIM total: $p < .001$, $\eta^2 p = .720$). No significant between-group differences or interaction effects were estimated.

As for the cognitive outcomes, significant improvements were found within the Group 1 in MoCA ($p = .04$) and TMT-A ($p = .02$) scores, while the Group 2 significantly improved SDMT ($p < .05$) scores. Significant main effects were also estimated on MoCA ($p = .03$, $\eta^2 p = .202$), SDMT ($p = .005$, $\eta^2 p = .332$), TMT-A ($p < .05$, $\eta^2 p = .182$), and FAB ($p = .02$, $\eta^2 p = .224$) scores. Again, no significant differences between the two groups or interaction effects were observed.

Regarding HRQoL and psychological variables, Group 1 significantly improved SF-12 PCS ($p = .002$) scores only. Differently, Group 2 showed significant changes in EQ-VAS ($p < .05$), SF-12 PCS ($p = .02$), GAD-7 ($p = .02$), and SAT-P resistance to physical fatigue ($p < .05$) and mental efficiency ($p = .05$) mean scores. Significant main effects were also estimated on EQ-VAS ($p = .005$, $\eta^2 p = .304$), SF-12 PCS ($p < .001$, $\eta^2 p = .647$), GAD-7 ($p = .003$, $\eta^2 p = .336$), PHQ-9 ($p = .02$, $\eta^2 p = .237$), and SAT-P mood ($p = .02$, $\eta^2 p = .233$) and resistance to physical fatigue ($p = .01$, $\eta^2 p = .268$). Further, at post-intervention evaluation, Group 2 reported significantly higher mean scores in SF-12 MCS ($p = .01$), GAD-7 ($p = .04$), and SAT-P mood ($p = .02$), resistance to physical fatigue ($p = .009$) and mental efficiency ($p = .009$) than the Group 1. Notably, a significant pre-post intervention between-group difference in SAT-P mental efficiency ($p = .02$) was estimated, along with a significant interaction between time and group ($p = .03$, $\eta^2 p = .200$).

Fig 4 shows the longitudinal changes over the three study timepoints. At 6-month follow-up, significant between-group differences were found in EQ-VAS ($p = .03$) and SF-12 PCS ($p = .04$) scores. Moreover, significant main ($p < .001$, $\eta^2 p = .467$) and interaction ($p = .02$, $\eta^2 p = .205$) effects were estimated on SF-12 PCS scores. Of these, a significant difference between baseline and 6-month follow-up scores was found within the Group 2 ($p = .005$), which reported significantly wider improvements than the Group 1 ($p = .04$). Similarly, a significant longitudinal difference was also found in PHQ-4 scores within both Group 1 ($p = .03$) and Group 2 ($p = .007$), along with a significant main effect of time ($p < 001$, $\eta^2 p = .372$).

## 4. Discussion

Adopting a prospective, two-arm, open-label, non-randomized design, the present study investigated the short- and long-term multi-domain effects of technology-enhanced neuromotor rehabilitation. Convenience sub-samples of patients with stroke, PD, or TKA/THA were drawn from a real-world neuromotor rehabilitation setting with the aim to provide pilot feasibility evidence on the perceived effectiveness of robotic and VR devices on multiple health outcomes, namely the functional status, cognitive functioning, HRQoL, and psychological status. Multidisciplinary rehabilitation programs, where part of the intervention was integrated with the implementation of technological devices, were delivered based on patients' individualized rehabilitation project and program. Pre- post-intervention changes and longitudinal effects over 6 months were analysed estimating intra- and inter-group effects. All analyses were conducted separately for each clinical sub-sample (i.e., stroke, PD, and OA), and the corresponding results were interpreted independently.

### 4.1. Stroke

Stroke patients reported widespread improvements after the intervention. In both study arms, patients reported significant changes in motor disability, but no group superiority was found. Regarding levels of independences in the ADLs, only those participating in technology-enhanced treatment showed significant improvements at post-intervention. Prior works [54], evaluating the use of robotics and VR to improve function in stroke survivors, underscored the potential of technology to increase therapy dose in terms of repetition and intensity, which is essential to achieve significant functional improvements in patients with acquired neurological conditions [55,56]. This may explain the differential effect found in terms of functional status within the two study groups. Despite no significant between-group differences emerged at post-intervention, the result suggested that integrating technology may have contributed to extending the rehabilitation effects to patients' level of autonomy beyond motor improvement.

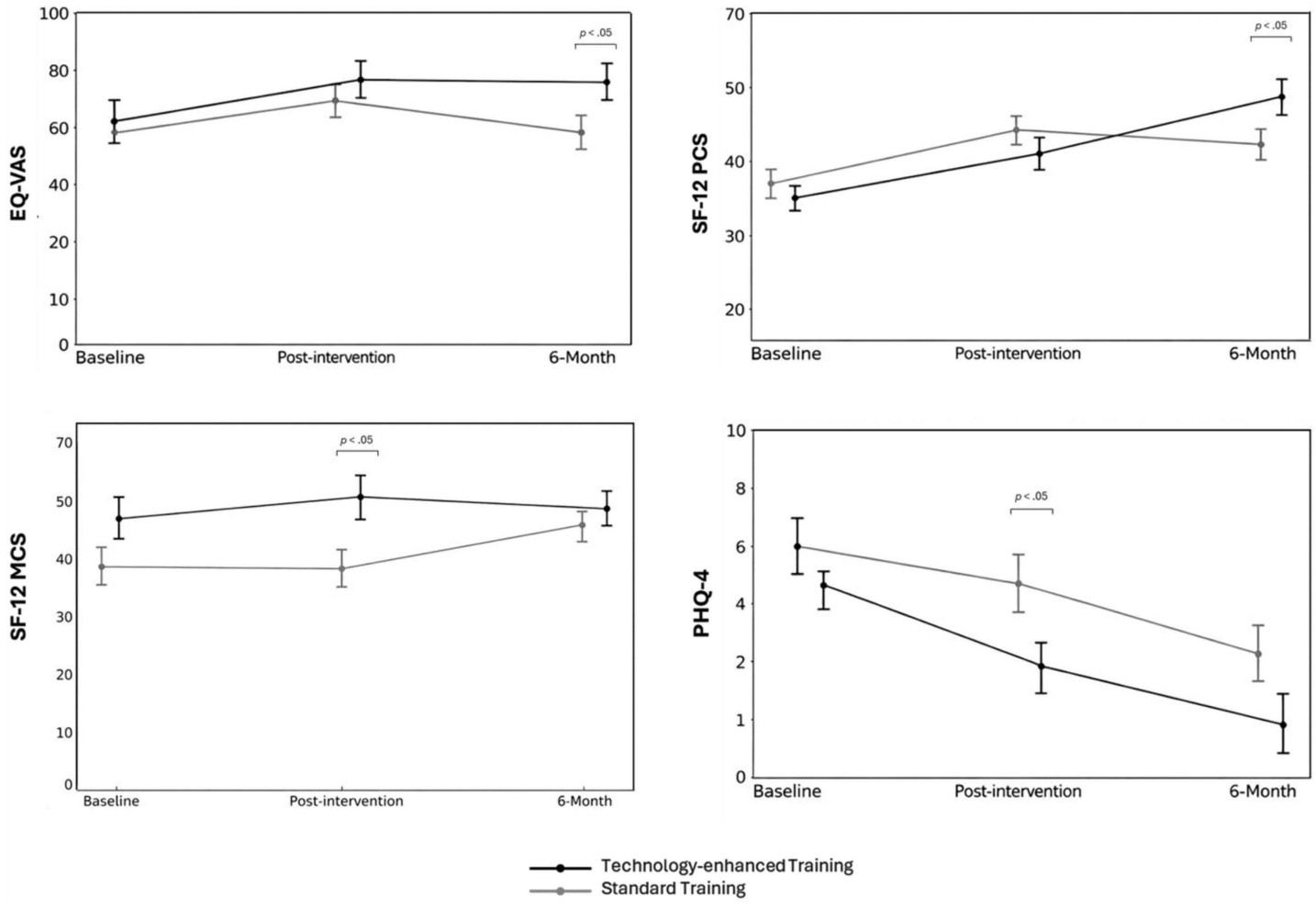

**Fig 4. HRQoL, anxiety and depression mean scores at baseline, post-intervention, and 6-month in patients with OA.**

Interestingly, significant changes were also found in the cognitive domains investigated. While the group participating in standard treatment showed significant adjustments in limited attentional skills, the group exposing to technology improved global cognition along with divided attention, cognitive flexibility, and executive functions. Standard approaches in stroke rehabilitation have widely evidenced the positive impact of motor recovery on cognitive domains, since sensory, motor, and cognitive aspects, sharing mutual neuronal circuits and processes, influence each other during training [57]. Notably, the extended effects found within the group undergoing technology-enhanced rehabilitation can be explained by the multimodal and multisensory stimuli patients were additionally exposed to. Consistently, recent evidence [58] supported the efficacy of the use of virtual environments combined with motion tracking systems, which contributed to wider post-stroke cognitive improvements when compared to usual care treatment. Likewise, robotic technology, especially when coupled to VR systems, was found to substantially impact cognition through augmented simultaneous motor and cognitive dual-task training [59,60], ultimately extending the effects of standard therapy. Overall, although no significant intergroup differences were estimated for cognitive change, again informative evidence on the integration of technology in stroke rehabilitation programs was provided by analysing the two study groups independently.

Informative results were also observed in HRQoL and psychological outcomes. Differently from the results found on the functional status and cognitive functions, significant post-intervention improvements were detected within both study groups regarding almost the same domains, including perceived overall health, QoL in terms of physical health and mental health, anxiety symptoms, and satisfaction with resistance to physical fatigue. Interestingly, only the patients undergoing technology-based training reported changes in relation to mood, with a significant decrease of depression symptoms and increase of perceived satisfaction with mood. Particularly for the latter, significant intergroup differences were also estimated, suggesting these patients to be more satisfied than those participating in standard treatment exclusively. In general, the evidence on the effects of technology-enhanced rehabilitation on HRQoL in stroke population, despite the most investigated so far, still remains mixed and unclear [25]. If on the one hand this makes it difficult to discuss and draw conclusions on technology effectiveness, on the other hand it may explain the lack of differential effects observed between the two study groups, including those estimated longitudinally at 6-month follow-up. Regarding the psychological outcomes, prior works with stroke patients [61–63] supported the positive impact of both standard and technology-enhanced programs on their emotional status and satisfaction with treatment following perceived motor benefits. This may explain the positive evidence observed among all study participants and the lack of significant intergroup effects. However, an exception can be found in the results on mood. As widely known, stroke can represent a clinical event profoundly impacting patients' participation and daily activities with the consequent onset of disease-related mood disorders, that standard rehabilitation programs often insufficiently address [14,56]. The integration of technological devices not only may have provided study participants with more engaging therapy sessions and better rehabilitation experience, with a possible contribution of novelty-related and expectancy effects on perceived well-being, but it has also significantly contributed to improve autonomy in ADLs and cognition that, in turn, may have positively affected mood and satisfaction with it.

### 4.2. Parkinson's Disease (PD)

Informative changes were also observed among patients with PD, despite the limited sample size at post-intervention study phase. Main effects of time were observed in the functional outcomes with significant intragroup improvements exclusively among patients participating in technology-enhanced programs. Besides positive changes in ADLs autonomy, these showed after the treatment significantly lower motor disability than those who underwent standard training. This result is in line with prior research suggesting both robotic and VR devices to be effective in improving motor and functional disability [7,64]. However, when compared to control interventions, the evidence supporting their added value is still controversial [8,65], ultimately suggesting that technology-enhanced rehabilitation have not always been superior to traditional physiotherapy in this clinical population. This may explain the lack of significant group effects observed in the present study.

Positive trends in cognition domains were also found. Although no significant changes after the intervention were estimated, patients of both study groups increased cognitive performance at neuropsychological assessment. Only a significant main effect of time in global cognition evaluation was observed, but no significant intergroup effects were detected. The lack of significant improvements can be due to different reasons. First, the limited sample size, especially regarding the group participating in standard rehabilitation, may have translated into reduced statistical power and wider variability, which in turn made it difficult to detect effects. Secondly, although the positive effects of neurorehabilitation programs on PD patients' cognition have been described [66], addressing such outcome remains an open challenge due to the disease complexity in terms of clinical progression. Nowadays, cognitive rehabilitation programs represent the most common therapeutic strategy to treat cognitive impairment in PD [67]. Notably, the present study sample participated in a multidisciplinary rehabilitation program, but none of them performed parallel cognitive stimulation activities. Moreover, other research on the cognitive effects of technology (e.g., VR) applied to physical rehabilitation provided mixed evidence with some studies reporting improvements in global cognition and executive functions [68,69] and others showing no significant results [70]. The findings of the present study seemed to be more in line with the latter.

Informative evidence was finally found in HRQoL and psychological outcomes. Differently from the patients undergoing standard therapy, those who participated in technology-enhanced programs reported significant changes in perceived overall health, depression symptoms, and satisfaction with mood, resistance to physical fatigue, and with mental efficiency. Despite no interaction effects emerged, significant between-group differences were found at post-intervention in particular with respect to perceived health and depression symptoms. These results are partially consistent with prior works that investigated rehabilitation technology effectiveness on non-motor outcomes. Of these, mixed evidence was reported with some studies showing HRQoL improvements after the use of technology when compared to control interventions [71–73] or not [74–76], and others estimating no significant effects [77–79]. The studies that concurrently investigated the impact on depression symptoms are less and, again, provided significant [73,80] or null results [80,81]. As for the improvements observed in satisfaction with mood, resistance with physical fatigue, and mental efficiency, this result represents an original finding, as it has not been reported yet, according to available literature. A possible explanation of these changes may be found in patient-reported rehabilitation experience, which for those participating in technology-enhanced programs was perceived as better, especially in terms of emotional support and physical comfort provided. However, the marginal statistical significance observed for physical comfort should be interpreted cautiously and warrants confirmation in larger trials, as it may partly reflect the limited samples size of this pilot study rather than a robust group difference. Overall, this inference can be supported by prior literature supporting the benefits of patient-centred rehabilitation contexts for multiple rehabilitation outcomes, including perceived health and psychological well-being [82–84].

In conclusion, the present study interestingly adds knowledge on the long-term impact of technology on HRQoL and anxiety and depression symptoms. A significant main effect of time was estimated for perceived physical health and a significant intergroup difference for anxiety and depression emerged at 6-month follow-up evaluation after patients' discharge. Despite preliminary, the present finding is original compared to the existing recent literature on patients with PD [77,78,85], which detected no significant longitudinal trends.

## 4.3. Osteoarthritis (OA)

Informative evidence was finally found among patients following TKA or THA. All patients involved in the study significantly improved motor disability and autonomy in the ADLs, but no intergroup effects were estimated at post-intervention, therefore suggesting that technology-based training was not superior in improving functional outcomes when compared to standard treatment. This finding may reflect the characteristics of contemporary post-arthroplasty rehabilitation, which increasingly relies on functional, task-oriented, and multi-joint training approaches rather than isolated joint strengthening [86], potentially limiting the additional motor benefits attributable to technology-assisted training. Furthermore, this result is partially in line with prior studies that supported the efficacy of VR rehabilitation technologies but, in some cases, lacked significant comparisons with control interventions [9,87]. Notably, most of these investigated physical and motor functions (i.e., gait, balance, range of motion, muscular strength, proprioception, and pain) as rehabilitation outcomes, providing poor insight into the effects on the related independence in the ADLs. Despite preliminary, the present study adds knowledge on patients' levels of disability and autonomy following technology-enhance rehabilitation program.

Post-intervention improvements were also observed in cognitive outcomes. Especially among patients participating in standard rehabilitation, significant changes in global cognition, attention, and executive functions were observed. Notably, no between-group differences were found. This result suggests that technology presumably had not a key role in enhancing study sample's cognitive profile, but rather the standard treatment mainly. In support of this, as already mentioned, standard physical rehabilitation can provide secondary effects on cognition beside motor improvements, as mutual neuronal circuits are activated during recovery [57]. Furthermore, the studies investigating the effects of VR-based rehabilitation on cognition in TKA/THA patients are scant, making it difficult to advance possible explanations about the role of this technology. Future studies should further investigate possible associations between the integration of VR devices in TKA/THA rehabilitation and cognitive improvement.

Regarding the results on HRQoL and the psychological outcomes, significant main effects of time were found in almost all the study variables. Significant between-group differences were estimated at post-intervention, with the patients participating in technology-enhanced programs reporting better scores in QoL in terms of mental health, anxiety symptoms, and satisfaction with mood, resistance to physical fatigue, and mental efficiency. Specifically for the latter, significant interaction effects were found, suggesting that technology-enhanced intervention has provided wider pre-post changes than standard treatment. The preliminary results on HRQoL are consistent with prior works showing positive effects of VR on perceived overall health [88] and physical health domain [89], differently from other works which found no significant effects [90,91]. As for the psychological outcomes, the significant difference found in anxiety levels is in line with prior recent works [92,93] that showed symptoms to be lower in patients after completing a VR-based intervention than in those who participated in standard treatment. In conclusion, the significant effects found in depression symptoms and satisfaction with mood, resistance to physical fatigue, and mental efficiency represent an original finding, as previous reviews and meta-analyses provided no related evidence [9,94,95]. Taken together, a possible explanation of the psychological effects observed may be found again in patient-reported rehabilitation experience. Patients participating in technology-enhanced programs remarked a more positive evaluation on the progress towards the achievement of rehabilitation goals, expectations, and performance areas that were important for them. This may reflect a positive perception of rehabilitation benefits that include both motor and non-motor functions.

The last finding concerns the longitudinal changes in HRQoL and anxiety and depression symptoms. At 6-month follow-up evaluation, significant between-group differences were found in perceived overall health and QoL physical health component, with the patients who participated in technology-based intervention reporting higher scores. As specifically regards physical health component, a significant interaction between time and study group was found, suggesting that technology has provided a more positive long-term impact than standard treatment. This finding is in contrast with a prior work [96] performing the same evaluation and that found no significant differences between the two study groups at 3-month follow-up. Regarding anxiety and depression symptoms, although no interaction effects emerged between the two study groups, both significantly decreased symptoms over 6 months. Based on previous recent reviews [9,94,95], no similar results have been reported in literature.

## 4.4. Study limitations and future directions

Overall, some limitations should be acknowledged from this study. Firstly, the sample sizes of all study groups are limited, especially in PD subsample, with consequent reduced stability of inferential estimates. According to guidelines for pilot studies, however, a convenience sample of at least six participants per group may be sufficient for investigating recovery outcomes in pilot rehabilitation trials that are at a feasibility stage [97]. Although satisfactory effect sizes were estimated, the results obtained cannot be considered as generalizable. Despite promising, the multi-sample evidence described so far should be taken as pilot and cautiously. Accordingly, given the exploratory and feasibility-oriented nature of the work, the inferential statistics reported was intended to describe clinical trends rather than support confirmatory conclusions. Further research with larger samples investigating the biopsychosocial effects of technology among the three pathologies involved in this study is strongly recommended.

Another limitation concerns the intervention characteristics and the absence of formal methods to quantify dose-response relationship. As already described, patients participated in different rehabilitation activities and experienced the use of different technological devices, at different dose, based on their individual rehabilitation project and program. Additionally, the non-randomized allocation based on routine multidisciplinary clinical decision-making and the grouping of heterogenous technological devices under a single intervention variable may have introduced potential selection bias and prognostic non-equivalence between study groups, thus limiting causal inference and findings generalizability. Although this contributed to obtaining greater intervention heterogeneity, the multidisciplinary and tailored approach adopted reflects the attempt to place the present study within a personalized medicine framework, whose adoption still represents one of

the major challenges in the field of rehabilitation medicine [3,29,98]. Moreover, this also aligns with the study's purpose of exploring intervention impact on a set of non-motor outcomes that are still insufficiently studied despite their recognized relevance. Nevertheless, it must be acknowledged that, in the present study, intervention variability may have introduced a confounding effect, sensibly limiting causal inferences. Moving forward, future studies should specifically address dose-response relationships while adopting controlled designs and systematically framing intervention intensity and exposure. Moreover, next research from the pilot evidence here reported may leverage the implementation of further cutting-edge tools (e.g., machine learning, artificial intelligence) to enhance training personalization and potentially enable a deeper analysis and understanding on non-motor mechanisms underlying technology-assisted recovery [99].

The last limit of this study may regard the use of self-report measure to investigate HRQoL and psychological outcomes. Despite self-report methods may be constrained by inferential limitations (e.g., social desirability bias, risk of false-positive), they are considered suitable to provide informative evidence of a phenomenon and have various advantages, including high practicality of use, clinical and research applicability, and good cost-effectiveness [100]. Moreover, the adoption of self-report evaluations strictly reflects the willingness to actualize the patient-centred approach adopted for the present study. Nonetheless, we acknowledge that relying exclusively on self-reported data may not fully capture individual changes. Similarly, the absence of structured cognitive stimulation within the rehabilitation programs may have limited the extent of cognitive improvements observed, which were not consistently significant across cognitive sub-domains. Future studies should consider including targeted cognitive training components and integrate more objective metrics.

## 5. Conclusions

Overall, this pilot study provides preliminary feasibility evidence supporting the integration of personalized, technology-enhanced neuromotor rehabilitation within multidisciplinary clinical pathways across heterogenous rehabilitation populations. Despite differences in etiology and clinical profiles, patients with stroke, PD and OA showed promising trends toward multi-domain benefits, extending beyond motor recovery to include functional autonomy, psychological well-being and HRQoL. While the exploratory nature and limited sample size warrant cautious interpretation, these findings highlight the potential added value of technology-enhanced and patient-centred rehabilitation approaches in real-world settings. Future adequately powered and controlled studies are needed to confirm these results, investigate dose-response relationships, and further clarify the mechanisms underlying multi-domain recovery.

## Supporting information

**S1 Appendix. Personalized intervention characteristics and 6-month follow-up complete analyses.**
(PDF)

**S2 Appendix. CONSORT-Checklist.**
(PDF)

**S3 Appendix. Copy of trial protocol summary approved by the Ethics Committee.**
(PDF)

## Author contributions

**Conceptualization:** Francesco Zanatta, Patrizia Steca, Anna Giardini, Antonia Pierobon.

**Data curation:** Francesco Zanatta, Cira Fundarò, Chiara Ferretti, Giovanni Arbasi, Antonia Pierobon.

**Formal analysis:** Francesco Zanatta, Roberta Adorni.

**Funding acquisition:** Antonia Pierobon.

**Investigation:** Francesco Zanatta, Cira Fundarò, Chiara Ferretti, Giovanni Arbasi, Antonia Pierobon.

**Methodology:** Francesco Zanatta, Roberta Adorni, Antonia Pierobon.

**Project administration:** Patrizia Steca, Cira Fundarò, Antonia Pierobon.

**Supervision:** Patrizia Steca, Anna Giardini, Marco D'Addario, Antonia Pierobon.

**Writing – original draft:** Francesco Zanatta.

**Writing – review & editing:** Francesco Zanatta, Patrizia Steca, Cira Fundarò, Anna Giardini, Chiara Ferretti, Giovanni Arbasi, Roberta Adorni, Marco D'Addario, Antonia Pierobon.

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
