## [Decision Letter · Decision Letter 0]

17 Jan 2025

Dear Dr. Fundarò,

Thank you for submitting your manuscript to PLOS ONE. After careful consideration, we feel that it has merit but does not fully meet PLOS ONE’s publication criteria as it currently stands. Therefore, we invite you to submit a revised version of the manuscript that addresses the points raised during the review process.

We look forward to receiving your revised manuscript.

Kind regards,

Joshua (Sung) H. You, PT, PhD

Academic Editor

PLOS ONE

Journal Requirements:

5. Please upload a copy of your study protocol that was approved by your ethics committee/IRB as a Supporting Information file. By the study protocol, we mean the complete and detailed plan for the conduct and analysis of the trial approved by the ethics committee/IRB. Please send this in the original language. If this is in a language other than English, please also provide a translation. [https://journals.plos.org/plosone/s/submission-guidelines#loc-guidelines-for-specific-study-types

Additional Editor Comments:

While The article, "Longitudinal impact of technology-enhanced patient-centered neuromotor rehabilitation on autonomy, cognition, quality of life, and psychological well-being: preliminary multi-sample evidence," offers a comprehensive and valuable contribution to the growing body of research in neuromotor rehabilitation, the following limitations and areas should be improved:

1. Sample Size and Generalizability:

o The study's small sample size across groups limits statistical power and external validity. While effect sizes are reported, a larger and more diverse sample would enhance the robustness of findings.

2. Non-Randomized Design:

o The non-randomized design introduces potential selection bias. Randomized controlled trials (RCTs) are needed to confirm the causal relationship between technology-enhanced interventions and observed outcomes.

3. Heterogeneity in Interventions:

o Participants received varying doses and types of technological interventions (e.g., VR versus exoskeleton). While this reflects real-world clinical practice, it complicates the attribution of effects to specific technologies.

4. Limited Cognitive and Psychological Insights:

o Cognitive outcomes, though promising, were not consistently significant. The lack of cognitive stimulation within the rehabilitation programs may have contributed to this limitation.

o Similarly, the psychological outcomes rely heavily on self-reported measures, which are prone to bias and may not fully capture nuanced changes.

5. Underreporting of Statistical Models:

o The article mentions using ANOVA and other statistical methods but lacks detailed reporting of model assumptions, adjustments, or corrections for multiple comparisons. This transparency is essential for reproducibility.

Reviewers' comments:

Reviewer's Responses to Questions

**Comments to the Author**

1. Is the manuscript technically sound, and do the data support the conclusions?

Reviewer #1: Partly

2. Has the statistical analysis been performed appropriately and rigorously?

Reviewer #1: No

3. Have the authors made all data underlying the findings in their manuscript fully available?

Reviewer #1: No

4. Is the manuscript presented in an intelligible fashion and written in standard English?

Reviewer #1: Yes

Reviewer #1: The present article describes a prospective two-arm, non-randomized pilot study to obtain multiple pieces of evidence on a multi-domain impact of technology-enhanced neuromotor rehabilitation from a convenience-based sample.

Said that the paper routinely presented p-values in all the comparisons. For a pilot study which is not a powered trial the meaning and presentation of such p-values does not make much sense. As such, the purpose of conducting a pilot study is to examine the feasibility of an approach that is intended to be used in a larger-scale study. A pilot study is not a hypothesis-testing study. Therefore, no inferential statistical tests should be proposed in a pilot study protocol, and any such tests conducted in a post-hoc fashion hold little practical meaning. Without inferential statistical tests, a pilot study will not provide p-values. Power analyses are used to determine the sample size needed to achieve adequate statistical power (typically 80% or 90%) for detecting a clinically meaningful difference with the specified inferential statistical test. However, power analyses should not be included in an application for a pilot study that does not propose inferential tests.

I recommend rewriting the paper to focus on it as a ‘pilot study’ aimed at evaluating the feasibility of recruitment, randomization, retention, assessment procedures, new methods, and the implementation of the novel intervention.

As a side note, since this is not a randomized trial, it is unclear how subjects are allocated to the two arms with nearly equal sample sizes. This process should be clearly described and further elucidated to understand its potential impact on a future randomized trial.

**Do you want your identity to be public for this peer review?** For information about this choice, including consent withdrawal, please see our Privacy Policy

Reviewer #1: No

---

## [Author Response · Author response to Decision Letter 1]

3 Jun 2025

Point-by-point response Letter to Editor and Reviewer [PONE-D-24-33691]

To the Editor and Reviewer,

Thank you for your time and the opportunity to revise our manuscript entitled: “Longitudinal impact of technology-enhanced patient-centered neuromotor rehabilitation on autonomy, cognition, quality of life, and psychological well-being: preliminary multi-sample evidence”.

Given that the revision process has taken several weeks, involving an editor reassignment and multiple clarification emails, we have included a summary of the key correspondence in this letter to provide context for the changes made and ensure full transparency.

In summary, before proceeding with the manuscript revision, we contacted the assigned Editor to clarify discrepancies that emerged from the initial editorial and reviewer comments. These discrepancies primarily concerned the framing of the manuscript as a ‘pilot study’, the appropriateness of including statistical reporting, and the need for clarification regarding outcome comparisons made within and between study subjects. In this letter, we report the key correspondence and explain how these issues were addressed and integrated into the revised manuscript.

Editor Comments:

While The article, "Longitudinal impact of technology-enhanced patient-centered neuromotor rehabilitation on autonomy, cognition, quality of life, and psychological well-being: preliminary multi-sample evidence," offers a comprehensive and valuable contribution to the growing body of research in neuromotor rehabilitation, the following limitations and areas should be improved:

1. Sample Size and Generalizability: The study's small sample size across groups limits statistical power and external validity. While effect sizes are reported, a larger and more diverse sample would enhance the robustness of findings.

2. Non-Randomized Design: The non-randomized design introduces potential selection bias. Randomized controlled trials (RCTs) are needed to confirm the causal relationship between technology-enhanced interventions and observed outcomes.

3. Heterogeneity in Interventions: Participants received varying doses and types of technological interventions (e.g., VR versus exoskeleton). While this reflects real-world clinical practice, it complicates the attribution of effects to specific technologies.

Limited Cognitive and Psychological Insights: Cognitive outcomes, though promising, were not consistently significant. The lack of cognitive stimulation within the rehabilitation programs may have contributed to this limitation. Similarly, the psychological outcomes rely heavily on self-reported measures, which are prone to bias and may not fully capture nuanced changes.

5. Underreporting of Statistical Models: The article mentions using ANOVA and other statistical methods but lacks detailed reporting of model assumptions, adjustments, or corrections for multiple comparisons. This transparency is essential for reproducibility.

Reviewer’s comments:

The present article describes a prospective two-arm, non-randomized pilot study to obtain multiple pieces of evidence on a multi-domain impact of technology-enhanced neuromotor rehabilitation from a convenience-based sample. Said that the paper routinely presented p-values in all the comparisons. For a pilot study which is not a powered trial the meaning and presentation of such p-values does not make much sense. As such, the purpose of conducting a pilot study is to examine the feasibility of an approach that is intended to be used in a larger-scale study. A pilot study is not a hypothesis-testing study. Therefore, no inferential statistical tests should be proposed in a pilot study protocol, and any such tests conducted in a post-hoc fashion hold little practical meaning. Without inferential statistical tests, a pilot study will not provide p-values. Power analyses are used to determine the sample size needed to achieve adequate statistical power (typically 80% or 90%) for detecting a clinically meaningful difference with the specified inferential statistical test. However, power analyses should not be included in an application for a pilot study that does not propose inferential tests. I recommend rewriting the paper to focus on it as a ‘pilot study’ aimed at evaluating the feasibility of recruitment, randomization, retention, assessment procedures, new methods, and the implementation of the novel intervention. As a side note, since this is not a randomized trial, it is unclear how subjects are allocated to the two arms with nearly equal sample sizes. This process should be clearly described and further elucidated to understand its potential impact on a future randomized trial.

Correspondence (highlights):

From: cira.fundaro@icsmaugeri.it

Sent: 2/4/2025

To: plosone@plos.org

Object: Clarification on Revision Procedures Following Reviewer and Editor Comments

Dear Editor,

We appreciate your time in handling our manuscript, "Longitudinal impact of technology-enhanced patient-centered neuromotor rehabilitation on autonomy, cognition, quality of life, and psychological well-being: preliminary multi-sample evidence."

After reviewing the comments from both the previous editor and the reviewer, we noticed some discrepancies that we would like to clarify before proceeding with our revisions. Specifically, while we acknowledge and accept the reviewer's suggestion to frame the study as a 'pilot study', we do not share the perspective that statistical results should be omitted entirely. The previous editor also emphasized the importance of transparent statistical reporting, which seems to contrast with the reviewer's recommendation to remove inferential statistical tests.

Could you please provide guidance on how we should reconcile these differing perspectives in our revision?

We appreciate your clarification and look forward to your advice.

Best regards,

Cira Fundarò

From: plosone@plos.org

Sent: 2/5/2025

To: cira.fundaro@icsmaugeri.it

Dear Dr. Fundarò,

Thank you for asking for more information on this comment. I have reached out to them for clarification and will be in touch when we have a response.

I appreciate your patience, and encourage you to reach out in one week if you haven’t heard back by then. If you have any questions in the meantime, please feel free to ask.

Best regards,

Roxanne Jastine Baltar Straive

Editorial Assistant

From: cira.fundaro@icsmaugeri.it

Sent: 2/13/2025

To: plosone@plos.org

Dear Dr. Baltar,

As suggested, I am following up on my previous email regarding the clarification on the discrepancies between the reviewer’s comments and the previous editor’s guidance. I wanted to check if there are any updates on this matter.

I appreciate your time and look forward to your response.

Best regards,

Cira Fundarò

From: cira.fundaro@icsmaugeri.it

Sent: 3/3/2025

To: plosone@plos.org

Dear Editor, I hope this email finds you well.

I am reaching out to follow up on any updates regarding the review process for our manuscript. Since today marks the deadline for the revisions, I wanted to check if there are any developments on this matter.

I would greatly appreciate any information you can provide at your earliest convenience. Thank you for your time and consideration.

Best regards,

Cira Fundarò

From: neurorehab@yonsei.ac.kr

Sent: 3/3/2025

To: cira.fundaro@icsmaugeri.it

Dear Authors,

It seems that your email may have been directed to the wrong recipient. As an Academic Editor, my role was to review your article and provide recommendations; however, the final decision rests with the Editor-in-Chief. I encourage you to reach out to them directly regarding the revision status and final decision.

That being said, a significant concern with your study lies in its design—particularly the comparison between subjects with neurological and musculoskeletal impairments, as well as the unequal sample size. A more rigorous approach would have been to compare subjects within a more homogeneous neurological population while maintaining a balanced sample size. This issue presents a substantial limitation, and I am uncertain whether it can be effectively mitigated or revised at this stage. However, other reviewers may have differing perspectives that support your current design. I am sorry that I can not support your current study design. I appreciate your efforts and encourage you to seek guidance from the Editor-in-Chief for further clarification.

Best regards,

Joshua

From: cira.fundaro@icsmaugeri.it

Sent: 3/3/2025

To: chenem25@gmail.com

Dear Dr. Emily Chenette, Ph.D.,

I hope this email finds you well. I am reaching out to request clarification regarding the status of our manuscript submitted to PLoS ONE entitled "Longitudinal impact of technology-enhanced patient-centered neuromotor rehabilitation on autonomy, cognition, quality of life, and psychological well-being: preliminary multi-sample evidence."

We initially sought guidance on how to reconcile the discrepancies between the reviewer’s comments and the previous editor’s recommendations, particularly regarding the statistical reporting. However, despite our repeated follow-ups over the past weeks, we have yet to receive clear guidance and were ultimately advised to contact you directly regarding the revision status and final decision. Given that today marks the deadline for submission, we would appreciate your guidance on how to proceed with our revisions to align with the journal's expectations. Specifically, we would like to understand how to address the conflicting recommendations, ensure our revision meets the required standards, and clarify the next steps in the process.

We would be grateful for your time and any direction you can provide.

Best regards,

Cira Fundarò

From: plosone@plos.org

Sent: 3/12/2025

To: cira.fundaro@icsmaugeri.it

Dear Dr. Fundarò,

Thank you for contacting us with this request, and thank you for submitting your work to PLOS One.

We have now received comments from the Academic Editor. Their comments are as follows:

"I fully support the transparent reporting of statistical results in the paper. However, I believe the reviewer's suggestion to remove the inferential statistical results may have stemmed from the fact that the study has a significant design flaw—namely, comparing two fundamentally different conditions: neurological impairment (apples) and musculoskeletal impairment (oranges). This inappropriate comparison raises concerns about the study’s validity from the outset."

Please note that you can always disagree with the comments provided by editors or reviewers, so long as you provide your reasoning for the disagreement in the Response to Reviewers file submitted alongside your manuscript. I hope this assists you in your revision process. If you have any further questions or concerns, please do not hesitate to let me or another staff member know.

Kind regards,

Marcelle Thomas

Publications Assistant

Point-by-point responses to Editor’s and Reviewer’s comments and revisions made

Editor Comments:

While The article, "Longitudinal impact of technology-enhanced patient-centered neuromotor rehabilitation on autonomy, cognition, quality of life, and psychological well-being: preliminary multi-sample evidence," offers a comprehensive and valuable contribution to the growing body of research in neuromotor rehabilitation, the following limitations and areas should be improved:

1. Sample Size and Generalizability: The study's small sample size across groups limits statistical power and external validity. While effect sizes are reported, a larger and more diverse sample would enhance the robustness of findings.

Overall, thank you for your note of appreciation regarding our contribution and for this first comment that allows us to better clarify that our main purpose, through this preliminary study, was to provide pilot evidence across three distinct and representative clinical populations undergoing neuromotor rehabilitation, specifically on a set of treatment outcomes that goes beyond motor functioning. We acknowledge that the small sample size across the three study groups may have limited findings robustness. Accordingly, we have mentioned this issue in “limitation” paragraph as follows:

“[…] the sample sizes of all study groups are limited. Although satisfactory effect sizes were estimated and adequate achieved statistical power was observed, the results obtained cannot be considered as generalizable. Despite promising, the multi-sample evidence described so far should be taken as pilot and cautiously.”

Additionally (also in accordance with reviewer’s suggestion), we have added to limitation section a literature reference regarding sample size appropriateness for pilot clinical trials, specifically conducted within motor rehabilitation settings, which supports that 6 participants are sufficient to pilot the method and the implementation of a novel intervention or approach and, subsequently, feed trial scale-up. Here we report the integration we have made:

“Overall, some limitations should be acknowledged from this study. Firstly, the sample sizes of all study groups are limited. According to guidelines for pilot studies, however, a convenience sample of at least six participants per group may be sufficient for investigating recovery outcomes in pilot rehabilitation trials that are at a feasibility stage [95]. […]”

Reference added: [95] Dobkin BH. Progressive Staging of Pilot Studies to Improve Phase III Trials for Motor Interventions. Neurorehabil Neural Repair. 2009;23(3):197-206. doi:10.1177/1545968309331863

2. Non-Randomized Design: The non-randomized design introduces potential selection bias. Randomized controlled trials (RCTs) are needed to confirm the causal relationship between technology-enhanced interventions and observed outcomes.

3. Heterogeneity in Interventions: Participants received varying doses and types of technological interventions (e.g., VR versus exoskeleton). While this reflects real-world clinical practice, it complicates the attribution of effects to specific technologies.

Thank you for these comments, which helped us to better articulate the rationale behind the intervention, reflecting routine clinical practices at the Institution where the trial was conducted. We acknowledge that varying doses and types of interventions may introduce considerable heterogeneity, making it difficult to directly attribute outcomes to specific intervention components. However, we support the value of the real-world data collected and analyzed in this study. As described in the “Intervention” section, rehabilitation treatment design and delivery were strictly individualized, based on each patient’s clinical diagnosis, disease severity, and specific rehabilitation needs. This personalized and patient-centred approach led to the development of tailored rehabilitation plans, which made participant randomization and intervention standardization unfeasible in the context of the present study. As previously mentioned, the primary aim at this phase was to pilot the effectiveness of this intervention approach - which integrates the ICD and ICF frameworks - by focusing on a set of non-motor outcomes that, according to the literature, warrant greater attention and remain underexplored. The study was conducted on a convenience sample. Here, we include intervention description and allocation modalities, and the related limitations acknowledged in discussion section (integration included in bold):

“Patients were assigned to the two study arms, depending on their rehabilitation program (Figure 1). This was defined following the routine multidisciplinary clinical practice of the Institution where the study was carried out [27]. In particular, each patient’s recovery pa

---

## [Decision Letter · Decision Letter 1]

29 Jan 2026

Dear Dr. Fundarò,

Thank you for submitting your manuscript to PLOS ONE. After careful consideration, we feel that it has merit but does not fully meet PLOS ONE’s publication criteria as it currently stands. Therefore, we invite you to submit a revised version of the manuscript that addresses the points raised during the review process.

We look forward to receiving your revised manuscript.

Kind regards,

Hiroki Annaka

Academic Editor

PLOS One

Journal Requirements:

Reviewers' comments:

Reviewer's Responses to Questions

**Comments to the Author**

Reviewer #2: All comments have been addressed

Reviewer #3: All comments have been addressed

Reviewer #4: All comments have been addressed

2. Is the manuscript technically sound, and do the data support the conclusions?

Reviewer #2: Yes

Reviewer #3: Yes

Reviewer #4: Partly

3. Has the statistical analysis been performed appropriately and rigorously?

Reviewer #2: Yes

Reviewer #3: Yes

Reviewer #4: N/A

4. Have the authors made all data underlying the findings in their manuscript fully available?

Reviewer #2: Yes

Reviewer #3: Yes

Reviewer #4: Yes

5. Is the manuscript presented in an intelligible fashion and written in standard English?

Reviewer #2: Yes

Reviewer #3: Yes

Reviewer #4: Yes

Reviewer #2: Comments to the authors:

General comments:

- All acronyms and abbreviations should be mentioned (example ADLs).

- The inclusion of OA is not a typical neuromotor disease??? The rationale of their inclusion needs to be stated?

Specific comments:

- The methods section of the abstract did not mention important details about what personalized technology enhanced neuromotor rehabilitation used for each group compared to standard training??

- Post intervention period for each subgroup??? Were all standardized at 4 weeks?

- What do you mean by interaction effects??? Line 40.

- Line 41: is not clear, evidencing wider short time effects???

- How this time effect was estimated?? Line 42.

Introduction:

- Extensive literature on the three pathological (diseases) conditions without providing details on how enhanced technology would alleviate consequences and complications, accelerate restoration of the multi-domain functionality, and improving the patient’s outcomes????

- The rationale for study conduction is not fully supported by the given background or the introduction????

- Previous literature on how such personalized technology-enhanced motor rehabilitation are scarce and inadequate.

- The aim of this pilot study was to provide feasibility evidence? Please elaborate???

- There is hypothesis (or hypotheses) for this longitudinal study???? Or the pilot

- Line 108 to line 118: this section should be part of the methods section with details and does not belong to the introduction or the background section.

Methods:

- Line 129: the time elapsed between study conduction and submission is almost three years!!! Authors should provide proper justification for such time gap???

- Again all acronyms and abbreviations should be mentioned (line THA, ----)

- Line 160 to line 163: it is a non-randomized pilot trial; this section is not needed?

- How the technology-enhanced group was assigned???

- Data collection: this section needs to use subtitle ---- allowing stating data collected at each stage for clarity.

- Line 190: all tests were adjusted for age and education status of the participants??? Elaborate please.

- Also, for tools used please refer to their validity and reliability (if possible).

- Never mention how the sample size was estimated and how the study power was calculated, what effect size used for each?

- Possible confounding and methods to adjust them were not given?

- Was there any method to quantify the does-response relationship, or at least recommend this future research direction.

Results:

- Table 1: what was group 1 and group 2??

- While within group, seems no statistical differences observed in the basic sociodemographics, and clinical characteristics, there were statistically significant differences between groups (age in PD) table 1???

- What is PIADs subscales??? Please mention the tool in the methods section??

- Physical comfort in table 2 with marginally significant p value could be due to small sample in this group rather than a real difference? Please justify.

- What would be the between group differences reporting PIADs?

- Figure 2: use whisker plot instead of the line chart with mean and errors, insert the P value in addition to whisker and plot.

- Figure 2: 4 panels including HRQoL, depression and anxiety??? Why 4???

- Figure 2: please provide label for each graph.

- Same comments for figure 3, PD.

Discussion:

- No need to describe the objective and methods in this section, please provide the answer to your research question.

- Although, conclusion for each condition was provided at the end of each, the grand conclusion for the whole study was not given at the end of the manuscript???

Reviewer #3: Longitudinal impact of technology-enhanced patient-centred neuromotor rehabilitation on autonomy, cognition, quality of life and psychological well-being: pilot multi-sample evidence

As the authors addressed the given comments, I suggest acceptance of the manuscript

Reviewer #4: General Comments

The manuscript investigates the effects of technology-enhanced neuromotor rehabilitation compared to standard care across three distinct pathologies (Stroke, Parkinson’s Disease, and Osteoarthritis) on non-motor outcomes. While the topic is relevant to current trends in rehabilitation, the study presents significant methodological flaws that undermine the validity of the conclusions drawn.

Major Weaknesses:

Sample Size and Statistical Power: The sample sizes, particularly for the Parkinson’s Disease (PD) subgroup (Control group $n=4$), are critically low. Conducting inferential statistics (ANOVA/t-tests) on a group of 4 participants against 8 participants yields results with extremely low statistical power and high susceptibility to Type I and II errors. The results for the PD subgroup are statistically fragile and likely unreliable.

Allocation Bias: The study uses a non-randomized, convenience sampling method where allocation was based on "resource allocation" and "routine clinical practice." This introduces severe selection bias. It is highly probable that patients assigned to the technology arm had different clinical profiles or prognoses (e.g., higher cognitive function or physical capability to handle the technology) than those assigned to standard care, despite the baseline comparison in Table 1.

Heterogeneity of Intervention: The "technology-enhanced" arm is highly heterogeneous. It combines robotics (Lokomat, Armeo-Spring) and Virtual Reality (ProKin, D-Wall) depending on the patient. Lumping these distinct modalities into a single "technology" variable obscures the specific effects of the interventions.

Control of Intensity: It is unclear if the metabolic demand and cognitive load were equated between the standard care and technology groups. If the technology group worked at a higher intensity (a common feature of robotic devices), the results may be due to intensity rather than the modality itself.

Minor Weaknesses:

Outcome Multiplicity: The study assesses a high number of dependent variables relative to the sample size. While Bonferroni corrections are mentioned, the risk of spurious significant findings remains high given the exploratory nature of the analysis.

Follow-up Attrition: There is a notable loss to follow-up (from $n=68$ enrolled to $n=52$ at 6 months). The analysis of the longitudinal data does not appear to adequately account for the characteristics of the dropouts.

Specific Comments

Abstract

Page 1, Line 57: The term "pilot feasibility evidence" is used here, but the conclusion (Line 79) claims "widespread effectiveness." These terms are contradictory. If this is a pilot feasibility study, effectiveness cannot be firmly established.

Page 1, Line 62: Provide numerical data (p-values or effect sizes) for the between-group differences rather than just listing the domains.

Page 1, Line 71: Clarify if the "significant time effects" at 6 months were sustained improvements or declines compared to post-intervention.

Introduction

Page 3, Line 95: The background section introduces emerging technologies but fails to adequately frame the current scope of motion analysis and wearable technology, which are the foundational mechanisms of the devices used in this study (e.g., TecnoBody). To better contextualize the "technology-enhanced" aspect, the introduction must reference recent scoping reviews that link these technologies to injury prevention and motion analysis.

Reference to add: [Souaifi M, Dhahbi W, Jebabli N, Ceylan Hİ, Boujabli M, Muntean RI, Dergaa I: Artificial intelligence in sports biomechanics: A scoping review on wearable technology, motion analysis, and injury prevention. Bioengineering 2025, 12(8):887.]

Page 4, Line 169: The discussion of "individualized recovery pathways" and "rehabilitation objectives" lacks a strong theoretical basis regarding the optimization of rehabilitation risks. The manuscript should acknowledge contemporary frameworks that integrate biomechanics with rehabilitation optimization to support the "personalized approach" claimed in the study design.

Reference to add: [Dhahbi W: Advancing biomechanics: enhancing sports performance, mitigating injury risks, and optimizing athlete rehabilitation. In., vol. 7: Frontiers Media SA; 2025: 1556024.]

Page 4, Line 171: The aim states to provide "evidence on the effectiveness." Given the non-randomized design and small sample, the aim should be downgraded to "exploring preliminary trends" or "assessing feasibility."

Page 5, Line 186: The hypothesis that technology participants would report "significantly wider multi-domain improvements" is ambitious given the heterogeneity of the chosen technologies.

Methods

Page 5, Line 206: Explicitly state the exclusion criteria regarding prior use of these technologies. Prior exposure could influence the "psychosocial impact" scores due to a lack of novelty effect.

Page 6, Line 251: The allocation method "resulted from Institution's routine effort to optimize resource allocation" is the critical methodological flaw. Define exactly how a clinician decided a patient received technology vs. standard care. Was it based on cognitive capacity? Physical severity? This implies the groups were not comparable by prognosis.

Page 7, Line 256: "Resulting in the same amount of therapy for all participants." Time-matched does not mean dose-matched. Provide data or a statement on how physiological or kinematic intensity was monitored to ensure equivalence.

Page 7, Line 258: Combining Lokomat (gait robot), Armeo (upper limb exoskeleton), and ProKin (balance board) into one independent variable is problematic. These target different motor systems. Acknowledge this limitation in the design section.

Page 8, Line 309: Specify if the assessor for the 6-month follow-up was blinded to the group allocation.

Results

Page 9, Table 1: The PD Control group has an $n=4$. Standard deviations and p-values derived from a sample of 4 are statistically unstable. The lack of significant difference at baseline ($p > 0.05$) does not prove equivalence when power is this low.

Page 11, Table 2: The statistical comparison of "Rehabilitation Experience" between groups is questionable given the open-label nature. Patients receiving novel, expensive technology (robots) are inherently likely to report higher satisfaction due to the perceived value of the care (placebo/expectancy effects) rather than the clinical efficacy.

Page 12, Table 3: The notation in this table is cluttered and difficult to interpret. Ensure that the "Between-group differences" column clearly indicates which group was superior.

Page 14, Line 490 (PD Results): The manuscript reports significant improvements in the PD technology group ($n=8$) but not the control ($n=4$). Attributing this to the intervention is flawed; it is equally likely that the control group was simply too small to detect the natural recovery or training effect that occurred.

Discussion

Page 16, Line 587: The statement "patients reported significant changes in motor disability" must be contextualized by the fact that both groups improved, and no between-group difference was found for motor disability.

Page 17, Line 620: The discussion on HRQoL improvements in Stroke lacks a critical evaluation of the "novelty effect" of technology on mood and perceived well-being, separate from physiological recovery.

Page 18, Line 651: The discussion of PD results must explicitly and repeatedly acknowledge the $n=4$ control group limitation. Discussing "effectiveness" or comparing these groups as if they were balanced cohorts is misleading.

Page 19, Line 704: The claim that these results "corroborate" the added value of providing comprehensive recovery is too strong. The findings are correlational at best due to the lack of randomization.

Page 20, Line 746 (OA Discussion): The analysis of motor function improvements in the OA group is superficial. It misses a critical comparison between the "focused" standard care provided and modern joint-by-joint training paradigms. This comparison is essential to understand why technology (which often engages multiple joints) might or might not offer superior outcomes compared to traditional strengthening.

Reference to add: [Dhahbi W, Materne O, Chamari K: Rethinking knee injury prevention strategies: joint-by-joint training approach paradigm versus traditional focused knee strengthening. Biology of Sport 2025, 42(4):59-65.]

Page 22, Line 828: The authors defend the heterogeneity as "personalized medicine." While clinically valid, in a research context without a crossover design or larger sample, this simply introduces noise. Rephrase this to acknowledge it as a confounder rather than a design strength.

Page 22, Line 831 (Future Directions): The suggestion for future research is generic. The conclusion ignores the specific potential of machine learning and deep learning applications in biomechanical analysis, which represents the next logical step for enhancing the efficacy and personalization of the technologies tested in this pilot study.

Reference to add: [Dhahbi W, Jebabli N, Boujabli M, Souaifi M, Dergaa I, Ben Ezzdine L: MACHINE LEARNING AND DEEP LEARNING APPLICATIONS IN SPORTS BIOMECHANICAL ANALYSIS: A SYSTEMATIC SCOPING REVIEW OF PERFORMANCE ENHANCEMENT AND INJURY PREVENTION STRATEGIES. ISBS Proceedings Archive 2025, 43(1):18.]

Conclusions

Page 2, Line 77: The conclusion "confirming the widespread effectiveness" is not supported by the data. The study confirms feasibility and suggests potential benefits, but "confirming effectiveness" requires a powered RCT. Rephrase to reflect the pilot nature of the data.

**Do you want your identity to be public for this peer review?** For information about this choice, including consent withdrawal, please see our Privacy Policy

Reviewer #2: **Yes:** Tarek Tawfik Amin

Reviewer #3: No

Reviewer #4: **Yes:** Wissem Dhahbi

---

## [Author Response · Author response to Decision Letter 2]

6 Feb 2026

Point-by-point response Letter to Editor and Reviewer [PONE-D-24-33691R1]

To the Editor and Reviewers,

We would like to thank you for your time and for the careful evaluation of our manuscript entitled: “Longitudinal impact of technology-enhanced patient-centered neuromotor rehabilitation on autonomy, cognition, quality of life, and psychological well-being: pilot multi-sample evidence”.

We are grateful for the constructive comments received during this second round of review. We are pleased that Reviewer #3 found the revised manuscript suitable for publication, and we sincerely thank the other Reviewers for their additional comments, which further helped us to improve the clarity, rigor, and overall quality of our work.

We have carefully addressed all remaining comments and suggestions. Revisions have been incorporated into the manuscript and, below, we provide a detailed point-by-point response to each comment. We hope that the revised version satisfactorily addresses all concerns and we thank you again for the opportunity to further enhance our research.

Reviewer #2:

General comments:

All acronyms and abbreviations should be mentioned (example ADLs).

Thank you for this suggestion. We have mentioned all acronyms and abbreviation when needed and consistently throughout the manuscript text.

The inclusion of OA is not a typical neuromotor disease??? The rationale of their inclusion needs to be stated?

Thank you for this valuable comment, which further helps us to better clarify the rationale behind the inclusion of OA patients in our study. We agree that OA is not a neurological condition. Its inclusion was basically motivated by the rehabilitation context rather than by disease etiology. Specifically, patients undergoing TKA/THA typically follow structured neuromotor rehabilitation programs aimed at recover motor control, balance, gait and autonomy. Within an ICF-based and biopsychosocial framework, OA therefore represents a musculoskeletal condition allowing us to explore the feasibility and multi-domain impact of technological rehabilitation across heterogeneous but representative clinical population. We have clarified this rationale in the Introduction paragraph (changes in bold):

“[…] For this purpose, a preliminary longitudinal investigation was conducted in convenience samples of patients drawn from three distinct but representative clinical populations typically undergoing neuromotor rehabilitation, namely stroke (neurological sub-acute), PD (chronic neurodegenerative), and OA (musculoskeletal). The latter was included as a representative musculoskeletal condition commonly undergoing structured neuromotor rehabilitation, particularly in the post-surgical phase”

Specific comments:

The methods section of the abstract did not mention important details about what personalized technology enhanced neuromotor rehabilitation used for each group compared to standard training??

Post intervention period for each subgroup??? Were all standardized at 4 weeks?

What do you mean by interaction effects???

Line 40. Line 41: is not clear, evidencing wider short time effects???

How this time effect was estimated?? Line 42.

Thank you for all these precious comments and suggestions on abstract’s details. Here we report the ‘Methods’ and ‘Results’ sections including the revisions made while preserving a concise abstract format (changes in bold):

Methods

A prospective, two-arm, non-randomized study design was adopted to provide pilot feasibility evidence on the multi-domain impact of personalized technology-enhanced neuromotor rehabilitation from convenience sub-samples of patients with stroke, Parkinson’s Disease (PD), and osteoarthritis (OA). Technological intervention consisted of the integrated use of robot-assisted and/or VR-based exercises, individualized based on patient’s diagnosis and rehabilitation goals. Study outcomes included patient’s functional status (autonomy in ADLs, risk of falls), cognition (attention and executive functions, memory, verbal fluency), physical and mental health-related quality of life (HRQoL), and psychological status (anxiety and depression symptoms, and well-being) and were compared to patients participating in standard individualized training only. Rehabilitation experience and technology psychosocial impact were also evaluated. Intra- and intergroup comparisons along with general linear models were statistically tested within each sub-sample considered independently over three timepoints (baseline, 4-week post-intervention, 6-month follow-up).

Results

At post-intervention, significant multi-domain intra-group improvements were observed within each sub-sample. Between-group differences were found on ADLs autonomy (stroke and PD), executive functions (stroke), anxiety and depression (OA and PD, respectively), and well-being (stroke and OA). Interaction effects (time x group) were significant only on well-being variables in stroke and OA, evidencing wider short-term effects of technology-enhanced programs compared to standard training. At 6-month, significant time effects over the three timepoints were estimated on HRQoL within each sub-sample and, additionally, on anxiety and depression in stroke and OA. Interaction effects emerged only on physical HRQoL in OA, along with significant between-group differences on HRQoL and anxiety and depression in OA and PD, respectively.

Introduction:

Extensive literature on the three pathological (diseases) conditions without providing details on how enhanced technology would alleviate consequences and complications, accelerate restoration of the multi-domain functionality, and improving the patient’s outcomes????

The rationale for study conduction is not fully supported by the given background or the introduction???? Previous literature on how such personalized technology-enhanced motor rehabilitation are scarce and inadequate.

Thank you for this important comment. We agree that the already revised version of the introduction placed emphasis on the clinical burden of the three conditions but lacks details on the specific mechanisms and advantages of technology-enhanced rehabilitation to support multi-domain recovery. Accordingly, we have revised the introduction to explicitly describe key benefits of robotic and VR-based training based on most recent literature on stroke, PD and OA. This revision was intended to strengthen the theoretical rationale while keeping a concise narrative and highlighting the existing gap in evidence on multi-domain rehabilitation outcomes (integration in bold):

“[…] Although the efficacy of technology to improve motor impairment has been supported widely [5-10], understanding the short- and long-term broader impact on patients’ non-motor characteristics still represents an open challenge [24]. Beyond motor recovery, emerging evidence suggests that technological devices may support multi-domain rehabilitation through several mechanisms, including the delivery of high-intensity and task-specific training, the provision of augmented multisensory feedback, and the facilitation of motor-cognitive integration. In post-stroke rehabilitation, robotic and VR-based interventions have been associated with improved motor learning and engagement, as well as potential secondary benefits on attention, executive functions, and mood [5-6]. In PD, technology-enhanced training has been shown to promote gait automatization, balance control, and dual-task performance, while also positively influencing motivation and perceived well-being [7-8]. Similarly, in OA and post-arthroplasty rehabilitation, VR-based interventions may enhance proprioception, balance, and functional mobility, while reducing risk and fear of falls and psychological distress [9-10]. Collectively, these advantages support the potential of rehabilitation technologies to restore functional limitations, mitigate secondary complications, and promote recovery across physical, cognitive, and psychological domains. Clearer evidence in this direction not only would extend knowledge on technology effectiveness to patient’s global functioning, but it would also corroborate the added value of providing comprehensive and technology-enhanced recovery programs […]”

The aim of this pilot study was to provide feasibility evidence? Please elaborate???

There is hypothesis (or hypotheses) for this longitudinal study???? Or the pilot

Thank you for these important comments which allowed us to better refine our study aims and hypotheses. By ‘multi-domain feasibility evidence’ we refer to the preliminary evaluation of the applicability and potential added value of individualized technology-enhanced neuromotor rehabilitation in real-world clinical settings, as well as its ability to capture changes across multiple outcome domains. Although the study was not designed to test confirmatory hypotheses, we had formulated priori exploratory expectations consistent with a pilot longitudinal design. Here we propose the following integration to better refine our research purposes (integrations in bold):

“Following this line, the present study aimed to provide pilot multi-domain feasibility evidence on the effectiveness of multidisciplinary, individualized, technology-enhanced neuromotor rehabilitation programs. Feasibility was intended as the preliminary assessment of the applicability and potential added value of technology-integrated rehabilitation within real-world clinical settings, as well as its sensitivity in capturing changes across multiple outcome domains. For this purpose, a preliminary longitudinal investigation was conducted […]. In line with the pilot and exploratory nature of the study, it was hypothesized that, compared to patients undergoing standard multidisciplinary rehabilitation alone, those participating also in technology-enhanced programs would show better short-term multi-domain trends and a more positive rehabilitation experience. It was further exploratorily expected that improvements in psychological well-being and HRQoL would be at least partially maintained at follow-up.”

Line 108 to line 118: this section should be part of the methods section with details and does not belong to the introduction or the background section.

Thank you for this comment. We acknowledge that this section includes methodological elements related to sampling and study outcomes. However, we preferred to retain it in the Introduction, as it was intended to support the study rationale by clarifying the choice of clinical populations and the real-world context in which the intervention was implemented. Detailed method information is fully reported in the Methods section. We believe that keeping this brief description in the Introduction helps reader better understand the conceptual framework and the objectives of the study.

Methods:

Line 129: the time elapsed between study conduction and submission is almost three years!!! Authors should provide proper justification for such time gap???

Thank you for raising this point. The time gap between study conduction and manuscript submission and revision was not due to delays in data analysis or reporting by the authors, but rather to external and organizational factors related to the clinical and institutional context where the study was carried out. Importantly, the study design, data collection procedures, and outcome measures remained consistent throughout this period, and the validity of the findings was not affected.

Again all acronyms and abbreviations should be mentioned (line THA, ----)

Thank you again for pointing this out. As already mentioned, we have mentioned all acronyms and abbreviations as appropriate throughout the manuscript text.

Line 160 to line 163: it is a non-randomized pilot trial; this section is not needed?

Thank you for this suggestion. This section was added from previous revision according to reviewer’s comment, that asked for clarifications concerning patient allocation in the absence of randomization. Given the non-randomized design, we considered this brief clarification important to ensure transparency and to contextualize the balance observed across study arms as a reflection of routine clinical practice rather than an allocation strategy. Accordingly, we preferred to retain this section.

How the technology-enhanced group was assigned???

Thank you for this comment. The assignment of patients to technology-enhanced group is described in detail in the “Intervention” paragraph. Briefly, patients were allocated to study arms based on their individualized rehabilitation programs and objectives, which were defined according to the Institution’s routine clinical practice. Specifically for technology-based program assignment, eligibility and its implementation followed rehabilitation technology-specific guidelines and recommendations. Here we report the extracts from the manuscript where these details are described:

“Patients were assigned to the two study arms, depending on their rehabilitation program and objectives (Figure 1). These were defined following the routine multidisciplinary clinical practice of the Institution where the study was carried out [27]. In particular, each patient’s recovery pathway was designed individually based on the integration between the International Classification of Diseases (ICD) and the International Classification of Functioning, Disability, and Health (ICF) models [28]. […] Only the patients assigned to the second study group arm underwent technology-based training. For these, part of the duration (which depended on rehabilitation objectives) of the whole intervention consisted of the use of exoskeletal (Lokomat®, Armeo-Spring® – Hocoma AG, Switzerland) and/or VR technology (ProKin, D-Wall, Walker View – TecnoBody SRL, Italy), resulting in the same amount of therapy for all participants. Technology-enhanced rehabilitation procedures, including eligibility criteria for assignment and implementation, followed appropriate rehabilitation technology-specific guidelines and recommendations [29–33].”

Data collection: this section needs to use subtitle ---- allowing stating data collected at each stage for clarity.

Thank you for this suggestion. We agree that clearly reporting timing of data collection is essential. In the present manuscript, however, we intentionally organized the section by outcome domains (functional, cognitive, psychological, and QoL measures), while explicitly reporting the timing of administration within each domain. We believe that this structure better reflects the multidimensional focus of the study and facilitates interpretation of the outcomes, while still ensuring clarity on assessment stages. For these reasons, we preferred to retain the current organization of the section.

Line 190: all tests were adjusted for age and education status of the participants??? Elaborate please.

Thank you for this precious request. We clarified that raw scores obtained from neuropsychological tests were adjusted for age and education to obtain standardized scores comparable with normative data. A brief clarification was added in “Data collection” section (in bold):

“Cognitive functioning was assessed through a comprehensive battery of neuropsychological tests. These included the Montreal Cognitive Assessment (MoCA) [39], which assessed global cognition (permission to use for research purposes was requested and training certification to administer the test was obtained). Attention, executive functions, and linguistic skills were further evaluated through the oral version of the Symbol Digit Modalities Test (SDMT) [40], the Trail Making Test (TMT-A and TMT-B) [41], the shortened version of the Stroop Colour Word test [42], the Frontal Assessment Battery [43], and the phonemic verbal fluency test [44] . All tests were adjusted for participants’ age and education to generate standardized scores comparable to normative data available for each test.”

Also, for tools used please refer to their validity and reliability (if possible).

Thank for this comment. To avoid overloading the Methods section with psychometric details, we did not report validity and re

---

## [Decision Letter · Decision Letter 2]

23 Feb 2026

Longitudinal Evidence of Technology-enhanced, Individualized Neuromotor Rehabilitation on Autonomy, Cognition, Quality of Life and Psychological Well-being: Pilot Multi-Sample Study

PONE-D-24-33691R2

Dear Dr. Cira Fundarò,

We’re pleased to inform you that your manuscript has been judged scientifically suitable for publication and will be formally accepted for publication once it meets all outstanding technical requirements.

Kind regards,

Hiroki Annaka

Academic Editor

PLOS One

Additional Editor Comments (optional):

Reviewers' comments:

Reviewer's Responses to Questions

**Comments to the Author**

Reviewer #4: All comments have been addressed

2. Is the manuscript technically sound, and do the data support the conclusions?

Reviewer #4: Yes

3. Has the statistical analysis been performed appropriately and rigorously?

Reviewer #4: Yes

4. Have the authors made all data underlying the findings in their manuscript fully available?

Reviewer #4: Yes

5. Is the manuscript presented in an intelligible fashion and written in standard English?

Reviewer #4: Yes

Reviewer #4: This pilot study investigates the multi-domain impact of technology-enhanced neuromotor rehabilitation across three distinct clinical populations: stroke, Parkinson’s Disease (PD), and osteoarthritis (OA). The revised manuscript effectively addresses the complexity of modern rehabilitation by shifting from a purely motor-centric focus to a biopsychosocial framework that includes cognitive and psychological outcomes. The inclusion of a 6-month longitudinal follow-up adds significant value to the feasibility evidence presented. The authors have successfully refined the text, clarified the non-randomized methodology, and provided a more robust discussion regarding the limitations of a pilot study with small convenience sub-samples

**Do you want your identity to be public for this peer review?** For information about this choice, including consent withdrawal, please see our Privacy Policy

Reviewer #4: **Yes:** Wissem Dhahbi

---

## [Editor Report · Acceptance letter]

PONE-D-24-33691R2

PLOS One

Dear Dr. Fundarò,

I'm pleased to inform you that your manuscript has been deemed suitable for publication in PLOS One. Congratulations! Your manuscript is now being handed over to our production team.

Kind regards,

on behalf of

Dr. Hiroki Annaka

Academic Editor

PLOS One